# Do LLM Agents Know How to Ground, Recover, and Assess? Evaluating Epistemic Competence in Information-Seeking Agents

**Jiaqi Shao**[1,2]**, Yuxiang Lin**[1]**, Munish Prasad Lohani**[1]**, Yufeng Miao**[3]**, Bing Luo**[1]*

[1]Duke Kunshan University
[2]The Hong Kong University of Science and Technology
[3]Microsoft AI

## Abstract

Recent work has explored training Large Language Model (LLM) search agents with reinforcement learning (RL) for open-domain question answering. However, most evaluations focus solely on final answer accuracy, overlooking how these agents reason with and act on external evidence. We introduce **SeekBench**, the first process-level evaluation framework for LLM search agents that operationalize *epistemic competence*. We develop and validate our annotation schema using an expert-annotated dataset of 190 traces (over 1,800 steps). To evaluate at scale, we introduce an LLM-as-judge pipeline. Our framework provides granular analysis of whether agents demonstrate: (1) **groundedness**, by generating reasoning steps supported by observed evidence; (2) **recovery**, by adaptively reformulating searches to recover from low-quality results; and (3) **calibration**, by correctly assessing whether current evidence is sufficient to provide an answer. By applying our evaluation framework to state-of-the-art search agents tuned on Qwen2.5-7B, we uncover critical behavioral gaps that answer-only metrics miss, as well as specialized skills such as Search-R1's synthesis abilities. These analyses highlight distinct epistemic competencies, offering insights for the development of more capable and trustworthy agents. Code is available at `https://github.com/SHAO-Jiaqi757/SeekBench`.

## 1 Introduction

Recent advances in Large Language Models (LLMs) have spurred a shift from models that require explicit prompting, such as chain-of-thought (Wei et al., 2022; Yao et al., 2023), toward autonomous LLM agents (Zhang et al., 2025a). These agents learn to solve complex tasks by optimizing a reasoning policy with reinforcement learning (RL), enabling them to implicitly learn decision-making strategies without requiring step-by-step external guidance (Jaech et al., 2024; Guo et al., 2025). Among these, *search agents* are developed to tackle information-seeking problems by enabling LLMs to interact with external knowledge sources and handle queries beyond training data (Zheng et al., 2025; Gao et al., 2025). Modern agents increasingly use richer toolboxes (e.g., page-level browsing, selective scraping, Python execution) yet they still manifest the same epistemic loop: identify an information gap $\rightarrow$ obtain external evidence $\rightarrow$ reason over it $\rightarrow$ decide the next action or final answer. Our framework targets this process, independent of the surface tool API, to maintain precise empirical scope. They process multi-turn reasoning, using external search tools, and integrating evidence (Xi et al., 2025a). The agent's responses produce multi-turn traces, which can be represented as:

$$\mathcal{T} = \langle \tau_1, \tau_2, \ldots, \tau_T \rangle, \tag{1}$$

where each turn $\tau_t$ consists of a tuple that can contain four possible steps: reasoning $r_t$, search $s_t$, evidence $e_t$, and answer $a_t$. Non-final turns ($t < T$) are of the form $\tau_t = \langle r_t, s_t, e_t \rangle$, while the final turn ($t = T$) concludes with an answer, *i.e.*, $\tau_T = \langle r_T, a_T \rangle$.

---

*Corresponding author

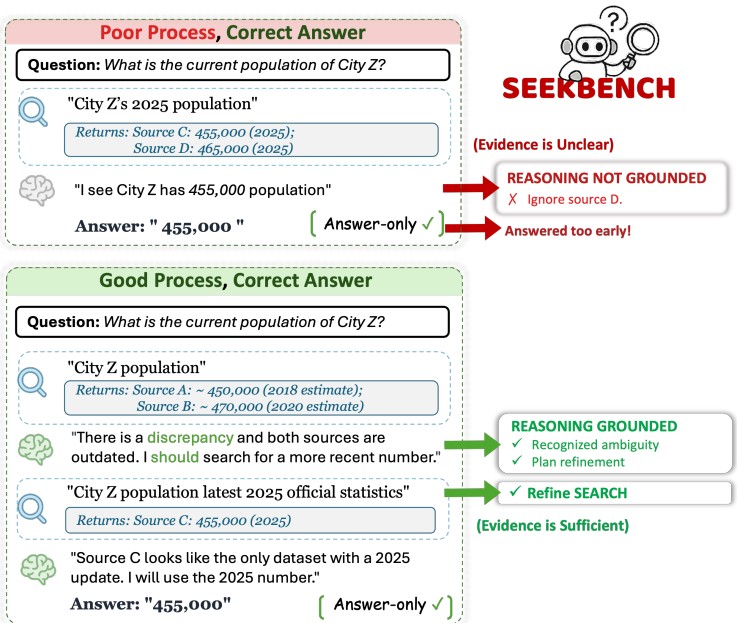

Figure 1: Contrasting accuracy with epistemic competence. Both scenarios yield the correct answer "455,000" but are evaluated differently by **SeekBench**. **Top (Poor Process):** Agent ignores conflicting sources, fails to recognize ambiguity, and answers prematurely without **grounding** reasoning or **recovering** from unclear evidence. **Bottom (Good Process):** Agent recognizes ambiguity, **refines** search to obtain recent official data, and *answers only after sufficient evidence*.

Information-seeking tasks pose a distinct epistemic challenge: unlike code generation or mathematical reasoning, where correctness can be externally verified by execution or proof (Nguyen et al., 2025; Lightman et al., 2023), search agents must reason about real-world textual information that lacks objective verifiers. However, current evaluation protocols for search agents largely ignore the process-level information captured by the structure of $\mathcal{T}$. They rely predominantly on answer-only ($a_T$) metrics such as exact match or F1 score (Jin et al., 2025; Zheng et al., 2025; Li et al., 2025b). This lack of process-level evaluation is problematic. As illustrated in Figure 1, *agents may achieve high benchmark scores while exhibiting poor epistemic behaviors*, such as hallucinating unsupported claims or failing to recognize knowledge gaps. This epistemic challenge necessitates process-level and evidence-driven evaluation to assess: **groundedness** (reasoning supported by evidence), **recovery** (adaptive search strategy to improve evidence), and **calibration** (determining when evidence is sufficient to answer).

To address the above challenges, we introduce **SeekBench** for evaluating agents' *epistemic competence*—the ability to reliably acquire, evaluate, and act upon knowledge in a justified manner (Greene et al., 2016; Zhang et al., 2023). Our framework proceeds in three stages. First, we developed a robust, extensible annotation schema for search agent traces that captures both the functional role (reasoning, search, answer) and epistemic quality (e.g., evidence groundness and sufficiency) of each step. Following established *Content Analysis* methodology (Krippendorff, 2018), we analyzed and annotated seven open-source search agents across 190 traces (over 1,800 steps, Figure 2; see Appendix C.1 for details). This expert-annotated dataset allowed us to validate and refine the schema, and the final schema achieves high inter-annotator agreement (Cohen's Kappa $\kappa > 0.8$). Second, we formalize three epistemic competencies: (1) **Groundedness** (evidence-grounded reasoning), (2) **Recovery** (adaptive evidence recovery), and (3) **Calibration** (evidence-aligned calibration). Third, we design precise, interpretable metrics to quantify these competencies across diverse traces, as detailed in Table 1. To evaluate at scale, we introduce an LLM-as-judge pipeline that enables scalable evaluation. This pipeline uses our validated annotation schema to automatically annotate agent traces and evaluate the competencies on a large scale.

**Contributions.** Our contributions are summarized as follows:

1. **SeekBench: A Process-Level Evaluation Framework.** We present **SeekBench** as a practical evaluation frameowork for empirical analysis of LLM information-seeking agents. **SeekBench** comprises a process-level annotation schema, operational epistemic metrics, and an automated

Table 1: Epistemic competencies and associated metrics. Each **competency** is quantified by a specific **metric** calculated from **annotated features** within the agent's trace (shown in the rightmost column), enabling systematic evaluation of reasoning quality, recovery behavior, and evidence-aligned decision-making.

| Competency (Type) | Definition & Metric | Annotated feature(s) |
|---|---|---|
| **Groundedness** (Reasoning) | Generate reasoning steps directly supported by retrieved information. **Metric:** *Reasoning Quality Index* (RQI, Section 3.3.1) | `InformationSynthesis` / `PlanFormation` / `StateAssessment` / `grounding` |
| **Recovery** (Search) | Adaptively reformulate queries when initial search results are insufficient. **Metric:** *Evidence Recovery Function* (ERF, Section 3.3.2) | `Initial` / `Repeat` / `FollowUp` / `Refined` |
| **Calibration** (Answer) | Accurately assess whether the currently retrieved information is sufficient to answer the question. **Metric:** *Calibration Error* (CE, Section 3.3.3) | `correct` (final answer's correctness) |

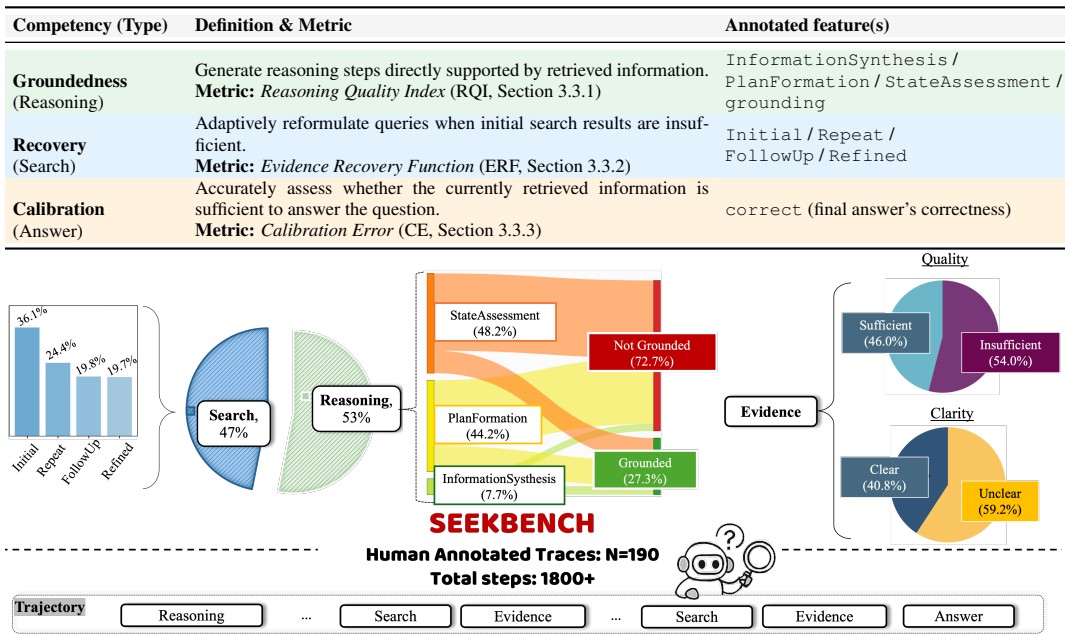

Figure 2: Overview of the **SeekBench** dataset and annotation schema. Each trace comprises multi-turn steps annotated for process-level evaluation. We categorize agent behaviors into three main types: (1) **Search steps** that retrieve information, (2) **Reasoning steps** that process evidence and guide the investigation, and (3) **Evidence steps** that capture the quality and clarity of retrieved information. This structured annotation approach enables systematic measurement of how well agents handle information throughout their reasoning process.

LLM-based judge pipeline. Our schema achieves high expert agreement (Cohen's Kappa $\kappa > 0.8$), and our LLM judge demonstrates strong alignment with expert annotations ($\kappa > 0.7$).

2. **Operational framework and metrics.** We formalize an *evidence state* and three core *epistemic competencies—evidence-grounded reasoning*, *evidence recovery*, and *calibrated answering*—as measurable properties over agent traces.

3. **Experimental case study.** We apply our framework to RL agents tuned on Qwen2.5-7B across seven QA benchmarks (28,493 traces), finding that they excel at gathering evidence but struggle with reasoning. Standard accuracy metrics fail to reveal strengths between agents (e.g., Search-R1's synthesis vs. Base model's reasoning), which can be combined to enhance performance.

## 2 RELATED WORK

**Process-Level Reasoning Quality and Epistemic Competence.** A fundamental limitation of answer-only evaluation is the disconnect between final-answer accuracy and reasoning process quality. Recent work on *faithfulness*, defined as logical consistency of reasoning with respect to questions and retrieved evidence (Lee & Hockenmaier, 2025), reveals that models can achieve high accuracy despite unfaithful reasoning processes (Shen et al., 2025). Other approaches evaluate reasoning quality through causal analysis of question-reasoning-answer triples (Paul et al., 2024), alignment with golden reasoning chains (Li et al., 2024), and graph-based dependency modeling (Xiong et al., 2025; Nguyen et al., 2024; Mukherjee et al., 2025). For search agents that interact with external information sources, epistemic competence becomes essential for avoiding overconfidence, hallucination, and poor decision-making under uncertainty. However, none of the existing frameworks can capture these capabilities. Our framework addresses this gap by formalizing three core epistemic competencies (groundedness, recovery, and calibration) with precise mathematical definitions and large-scale evaluation protocols.

**Search Agent Evaluation.** Existing evaluations primarily prioritize final-answer metrics (exact match, F1, LLM-as-Judge) (Zhang et al., 2025b; Song et al., 2025; Zheng et al., 2025; Jin et al., 2025), neglecting the underlying epistemic processes. While some approaches examine intermediate steps—such as ground-truth tracking (Shi et al., 2025) or retrieval separation (Xi et al., 2025b)—they

fail to assess critical epistemic competencies. Final-answer metrics also fail to attribute performance sources, as (Shao et al., 2025) finds that RL-training tends to elicit existing reasoning behaviors rather than introducing new ones. This methodological limitation creates significant blind spots regarding agent reliability.

# 3 METHODOLOGY

To understand the information-seeking process of search agents, our framework connects observable behaviors with underlying competencies, which are then evaluated using quantitative metrics. First, we construct and validate an **annotation schema** that reliably labels observable behaviors in agent response traces, producing the expert-annotated dataset that we also use to calibrate LLM judges (Section 3.1). Second, we analyze these annotations to identify three **fundamental epistemic competencies** (Section 3.2). Finally, we translate the competency definitions into concrete **quantitative metrics** and apply them to annotated features, enabling systematic measurement of these underlying epistemic competencies at scale (Section 3.3). Our approach draws on established qualitative research principles from *Content Analysis* (Krippendorff, 2018), a systematic methodology for categorizing and interpreting patterns in data through rigorous coding procedures.

## 3.1 PHASE 1: OBSERVABLE FEATURES AND SCHEMA CONSTRUCTION

The foundation of our methodology is a robust annotation schema for systematically labeling *observable features* in agent response traces. Our development of **SeekBench**'s schema follows an iterative, data-driven approach, grounded in established qualitative research principles from *Content Analysis* (Krippendorff, 2018).

During the initial exploratory phase, we closely examined a variety of agent traces and documented the key behaviors we observed. We noted that even within the same type of steps, search agents can serve distinct **functions**, that is, specific cognitive or operational role that a step plays within the agent's information-seeking process. For example, among the reasoning steps, some identified information gaps, while others summarized retrieved findings or formulated plans for future searches. Similarly, search steps might function as initial exploration, targeted verification, or follow-up investigation. Alongside these functions, we identified critical **failure patterns** in agents' traces, such as reasoning without supporting evidence or executing repetitive search queries that failed to adapt.

From these observations, we developed an annotation schema (detailed in Appendix A's Figure 9) for agent response steps with two key aspects:

- **Functional type** categorizing the step's cognitive purpose, e.g., for reasoning steps, this includes `InformationSynthesis` (evidence integration), `PlanFormation` (search strategy development), and `StateAssessment` (knowledge gap identification).
- **Quality attribute** evaluating epistemic soundness, such as whether reasoning is `grounded` in evidence. This structure captures both what the agent was doing and how well it was doing it.

We apply this schema to construct an expert-annotated validation dataset of 190 traces (Figure 2), which enables us to formalize three epistemic competencies and design quantitative metrics that systematically measure how annotated features map to underlying competencies (Table 1).

Following established *Content Analysis* methodology (Krippendorff, 2018), we rigorously enhanced schema robustness through *iterative refinement*. Using a setup comparable to previous works scaling from human to LLM-based annotation (Cemri et al., 2025), three expert annotators independently coded 190 agent traces across three rounds of annotation, with inter-annotator reliability measured using Cohen's Kappa ($\kappa$) (Cohen, 1960). For features exhibiting low agreement ($\kappa < 0.5$), we either pruned features (when infrequent or ambiguous) or merged them (when conceptually overlapping). After this process, we reduced our initial 12 candidate annotation fields to 8 well-defined features with high interpretability and consistency across annotators. We confirm this schema generalizes to complex agents in Appendix M.

We further validated schema robustness using GPT-5 (OpenAI, 2025) to generate adversarial edge cases that expose boundary conditions (e.g., reasoning steps containing both factual claims and planning elements). This ensures the mutual exclusivity of our annotation definitions.

Table 2: Examples of evidence states. $E = C + Q$, where $C \in \{0, 1\}$ indicates clarity and $Q \in \{0, 1\}$ indicates sufficiency.

| **Question:** *Who is the singer for the band Black Sabbath?* | | | | |
|---|---|---|---|---|
| $E$ | $C$ | $Q$ | **Search Result** | **Explanation** |
| **0** (Poor) | **0** | **0** | **Doc 1:** The band's **vocal slot has seen many**. 
 **Doc 2:** Black Sabbath's **personnel list for vocals** is long. | $C = 0$: text is vague and evasive. 
 $Q = 0$: no names are given. |
| **1** (Partial) | **1** | **0** | **Doc 1:** Black Sabbath is a famous **heavy metal band** from England. 
 **Doc 2:** Tony Iommi is the **guitar player** for Black Sabbath. | $C = 1$: on-topic and easy to understand. 
 $Q = 0$: does not name the singer. |
| **1** (Partial) | **0** | **1** | **Doc 1:** The first singer for Black Sabbath was **Ozzy Osbourne**. 
 **Doc 2:** **Ronnie James Dio** became the new singer for Black Sabbath. | $C = 0$: two singer names creates ambiguity. 
 $Q = 1$: the names are present. |
| **2** (Good) | **1** | **1** | **Doc 1:** **Ozzy Osbourne** is the original lead singer of Black Sabbath. 
 **Doc 2:** The most famous singer for Black Sabbath is **Ozzy Osbourne**. | $C = 1$: the name is consistent. 
 $Q = 1$: the singer is identified. |

Finally, we evaluated the schema on reasoning traces using both human experts and state-of-the-art LLM judges with standardized prompts that provide clear annotation guidelines and consistent evaluation criteria (see Appendix D). The results demonstrate substantial agreement with human annotations (overall $\kappa = 0.811$), confirming the schema's interpretability and consistency. LLM judges achieved strong alignment with human experts: GPT-4.1-mini ($\kappa = 0.731$) and GPT-5 ($\kappa = 0.754$). We further conducted a cost-effectiveness analysis, evaluating per-trace costs (token and time) across six LLM models. GPT-4.1-mini emerges as the most cost-effective solution, achieving strong human alignment ($\kappa = 0.731$) at minimal per-trace cost ($0.0087 and 2.48s), making it optimal for large-scale deployment (see Appendix A for cost analysis details). This substantial agreement across multiple LLM models and human annotators confirms the schema's clarity and establishes a reliable foundation for large-scale evaluation through LLM judges.

## 3.2 PHASE 2: LATENT CONSTRUCTS AND COMPETENCY DEFINITION

Our annotation analysis in Phase 1 revealed three distinct behavioral patterns in search agents: (1) variation in reasoning quality, with successful agents producing evidence-supported `grounded` reasoning while unsuccessful agents generated unsupported claims; (2) divergent strategies when facing poor search results, where effective agents adapted their search approach while ineffective agents persisted with repetitive queries; and (3) differences in decision timing, where some agents responded prematurely with insufficient evidence while others appropriately withheld answers until sufficient evidence was gathered. To interpret these systematic behavioral differences, we apply *latent construct inference* (Cronbach & Meehl, 1955) and arrive at three theory-backed competencies that explain the observed patterns (Table 1): (1) **Groundedness** captures whether each reasoning step is faithful to retrieved evidence, extending faithfulness checks from final answers to intermediate traces (Lee & Hockenmaier, 2025); (2) **Recovery** measures how effectively an agent reformulates and adapts its search to move from insufficient to sufficient evidence, mirroring information-foraging dynamics between exploration and query refinement (Pirolli & Card, 2005); (3) **Calibration** evaluates whether answer timing aligns with evidence sufficiency, drawing on metamemory research about calibrated confidence and appropriate abstention (Nelson, 1990; Ming et al., 2024). These three core competencies constitute **epistemic competence**—the essential capability that enables search agents to reliably interact with external information sources. By systematically evaluating how agents seek, reason with, and make decisions based on retrieved evidence, our framework provides a *comprehensive* assessment of search agents' epistemic capabilities beyond traditional accuracy-based metrics.

## 3.3 PHASE 3: COMPETENCY METRICS AND OPERATIONALIZATION

Following the concept of *construct validity* (Cronbach & Meehl, 1955) originally proposed in psychology, unobservable attributes (competencies) must be assessed through observable indicators (metrics) with demonstrated reliability and validity. In this section, we translated the three epistemic competencies in Phase 2 (defined in Table 1) into quantitative metrics. The validity of our metrics is twofold: (1) high inter-annotator agreement on the coded features (Phase 1), and (2) the correlation between evidence state (defined below) and answer accuracy (Section 4.4).

We begin by formally defining **evidence state** in Definition 3.1, which encodes the sufficiency and clarity of evidence retrieved at a turn (examples in Table 2). This provides the foundation for eval-

uating all three epistemic competencies: (i) **groundedness** is assessed by determining whether reasoning is supported by evidence (Section 3.3.1); (ii) **recovery** is measured by tracking improvements in evidence quality through search (Section 3.3.2); and (iii) **calibration** is evaluated on whether the agent answers if and only if the evidence state is good (Section 3.3.3).

**Definition 3.1 (Evidence State)** *Let $C_{i,t}, Q_{i,t} \in \{0,1\}$ denote the annotated* `clarity` *and* `quality` *(Appendix D.3) of the retrieved evidence at turn $t$ of trace $i$, where: $C_{i,t} = 1$ if the evidence is clear (unambiguous and interpretable), and $Q_{i,t} = 1$ if the evidence is sufficient (contains enough information to address the query). Note that* `quality` *here specifically refers to **sufficiency** (whether the evidence contains enough information to answer the query), not a general quality assessment. The **evidence state** $E_{i,t} \in \{0,1,2\}$ is defined as:*

$$E_{i,t} := C_{i,t} + Q_{i,t}, \tag{2}$$

*$E_{i,t} = 0$ denotes **poor** evidence (unclear and insufficient), $E_{i,t} = 1$ denotes **partial** evidence (either clear or sufficient), and $E_{i,t} = 2$ denotes **good** evidence (both clear and sufficient).*

### 3.3.1 GROUNDEDNESS

To evaluate whether an agent's reasoning is verifiably supported by retrieved evidence, we assess the **groundedness** of each reasoning step via the `grounding` label. For each reasoning step at turn $t$ in trace $i$, the binary `grounding` label $G_{i,t} \in \{0,1\}$ indicates whether its factual content is supported by retrieved evidence.

To investigate the impact of the functional types of the reasoning steps, each reasoning step is also assigned a `type` $\mathcal{C}_{i,t} \in \{IS, PF, SA\}$, corresponding to `InformationSynthesis`, `PlanFormation`, or `StateAssessment` (Appendix D.1).

We formalize two metrics: the **model-level reasoning quality index (RQI)** (Definition 3.2) and the **type-level RQI** (Definition 3.3), both of which quantify groundedness by aggregating $G_{i,t}$ values and can be decomposed by the evidence state $E_{i,t}$.

**Definition 3.2 (Model-level Reasoning Quality Index (RQI))** *Consider a fixed model evaluated on $N$ traces with index set $\mathcal{I} := \{1, \ldots, N\}$. For each trace $i$, let $S_i = \{1, \ldots, T_i\}$ be the index set of reasoning steps. Then, the model-level RQI is the average of trace-level groundedness scores:*

$$\text{RQI}_{\text{model}} := \mathbb{E}_{i \in \mathcal{I}}[\text{RQI}_i]. \tag{3}$$

*where $\text{RQI}_i = \mathbb{E}_{t \in S_i}[G_{i,t}]$,*

To better understand how reasoning quality depends on the strength of retrieved evidence, we decompose the RQI with evidence state $E_{i,t}$:

$$\text{RQI}_i = \sum_{k=0}^{2} \underbrace{\mathbb{P}_{t \in S_i}(E_{i,t} = k)}_{\text{proportion of turns with evidence state } k} \times \underbrace{\mathbb{E}_{t \in S_i}[G_{i,t} \mid E_{i,t} = k]}_{\text{expected groundedness given } E_{i,t} = k} \tag{4}$$

Similarly, we can define the type-level RQI with reasoning type $c \in \{IS, PF, SA\}$:

**Definition 3.3 (Type-Level Reasoning Quality Index)** *For each trace $i$ and reasoning type $c \in IS, PF, SA$, let $S_i^{(c)} := \{t \in S_i : \mathcal{C}_{i,t} = c\}$ denote the index set of steps of type $c$. The type-level RQI is the average of groundedness on type $c$:*

$$\text{RQI}_{\text{type}}^{(c)} := \mathbb{E}_{t \in S_i^{(c)}}\left[\text{RQI}_i^{(c)}\right], \tag{5}$$

*where $\text{RQI}_i^{(c)} := \mathbb{E}_{t \in S_i^{(c)}}[G_{i,t}]$. This quantity admits an evidence-state decomposition analogous to the trace-level decomposition:*

$$\text{RQI}_i^{(c)} = \sum_{k=0}^{2} \underbrace{\mathbb{P}_{t \in S_i^{(c)}}(E_{i,t} = k)}_{\text{prop. of reasoning type } c \text{ with evidence level } k} \times \underbrace{\mathbb{E}_{t \in S_i^{(c)}}[G_{i,t} \mid E_{i,t} = k]}_{\text{expected type } c \text{ groundedness given } E_{i,t} = k} . \tag{6}$$

### 3.3.2 RECOVERY

A fundamental challenge for LLM-based agents is recovering from information gaps or knowledge limitations, where initial queries yield insufficient information. Thus, an agent achieves high **recovery** when it utilizes adaptive search strategies to quickly escape such states of poor evidence.

To capture this behavior, we use the evidence state $E_{i,t} \in \{0, 1, 2\}$ to track the sufficiency and clarity of retrieved information at each turn $t$ in trace $i$. We define a *recovery event* (Equation (7)) as the first turn where the agent either (i) enters a high-evidence state ($E_{i,t} = 2$), or (ii) produces a correct answer. Formally:

$$T_{\text{recover},i} := \min\{t \in [1, T_i] : E_{i,t} = 2 \text{ or } \texttt{correct}_i = 1\}, \tag{7}$$

where $\texttt{correct}_i$ indicates whether the agent's final answer in trace $i$ is correct.

To measure recovery behavior, we introduce the Evidence Recovery Function (ERF), which quantifies the cumulative proportion of traces that have successfully recovered by each turn:

**Definition 3.4 (Evidence Recovery Function (ERF))** *Let $N$ denote the total number of traces. The* Evidence Recovery Function *at turn $t$ is defined as*

$$\text{ERF}(t) := \frac{1}{N} \sum_{i=1}^{N} \mathbb{I}\left(T_{recover,i} \leq t\right), \tag{8}$$

*where $\mathbb{I}(\cdot)$ is the indicator function. $\text{ERF}(t)$ measures the proportion of traces that have recovered by turn $t$.*

### 3.3.3 CALIBRATION

We evaluate **calibration** as the agent's ability to decide *when to answer* based on the quality of retrieved evidence. A well-calibrated agent should answer only when it has acquired evidence that is both clear (unambiguous and directly relevant) and sufficient (contains enough information to support a reliable answer).

Let $\texttt{answer}_{i,t} \in \{0, 1\}$ indicate whether the agent provides an answer at turn $t$ of trace $i$. We assess calibration behavior by examining the answer rate conditioned on the evidence state:

$$\mathbb{P}(\texttt{answer}_{i,t} = 1 \mid E_{i,t} = k). \tag{9}$$

High values at $k = 0$ indicate *epistemic overconfidence*, where the agent answers prematurely with poor or partial evidence. Conversely, low values at $k = 2$ suggest *epistemic overcautiousness*, where the agent refrains from answering when the evidence is good.

To quantify calibration performance, we introduce **Calibration Error (CE)** that measures how much an agent's answering behavior deviates from the ideal policy. The ideal policy is one that answers if and only if the evidence is good ($E_{i,t} = 2$), which maximizes expected accuracy while minimizing wasted effort. This metric captures both epistemic failures: overconfidence (answering with insufficient evidence) and overcautiousness (not answering despite having good evidence).

**Definition 3.5 (Calibration Error (CE))** *Let $\mathcal{I} := \{1, \ldots, N\}$ be the index set of traces. Let $\pi^*(k) := \mathbb{I}[k = 2]$ represent the ideal policy that answers if and only if evidence is sufficient. The CE for a model is defined as:*

$$\text{CE} := \mathbb{E}_{i \in \mathcal{I}}[\text{CE}_i] \tag{10}$$

*where for each trace $i$, $\text{CE}_i := \sum_{k=0}^{2} \mathbb{P}(E_{i,t} = k)\left|\mathbb{P}(\texttt{answer}_{i,t} = 1 \mid E_{i,t} = k) - \pi^*(k)\right|$.*

For a perfectly calibrated agent following the ideal policy $\pi^*(k)$, it achieves $\text{CE} = 0$.

## 4 EXPERIMENTS

### 4.1 EXPERIMENTAL SETUP

**Models & Datasets** . We evaluate Qwen-2.5-7B-Instruct ("Base" for training, Qwen et al. (2024)), its few-shot prompted version ("Few-shot"), and state-of-the-art "RL-trained" agents based on Qwen-2.5-7B-Instruct, including: SEARCH-R1 (Jin et al., 2025), RESEARCH (Chen et al., 2025), ASEARCHER (Gao et al., 2025) and DEEPRESEARCHER (Zheng et al., 2025). We also evaluate Chain-of-Thought (CoT) and ReAct prompting strategies, which show similar performance to Base (see Appendix F-I for details).

We evaluate the agents on a diverse set of seven QA benchmarks: NQ (Kwiatkowski et al., 2019), TriviaQA (Joshi et al., 2017), and PopQA (Mallen et al., 2022) [single-hop]; HotpotQA (Yang et al.,

Figure 3: **RQI Analysis Summary.** *Left:* RQI by model level, showing the overall reasoning quality across different agent types. *Right:* RQI by reasoning type, revealing that models struggle most with plan formation and state assessment compared to information synthesis.

2018), 2Wiki (Ho et al., 2020), MusiQue (Trivedi et al., 2022), and Bamboogle (Press et al., 2022) [multi-hop]. Additionally, we evaluate search agents (32B ASearcher and WebSailor) on GAIA to assess epistemic competencies with web browsing capabilities (see Appendix K for details).

Each model runs and evaluates on the **sanitized** test datasets, where we remove ambiguous questions and data contamination cases to ensure evaluation quality; we also removed questions that can be answered by the internal knowledge of the evaluated models alone since **evidence state** (Section 3.3) should depend only on retrieved evidence (details for data sanitization in Appendix B). Our evaluation comprises **28,493** traces and **283,950** steps across all models and datasets. A comprehensive statistical analysis of the dataset composition and annotation distributions is provided in Appendix C. For this large-scale annotation, we employ GPT-4.1-mini, which demonstrates substantial alignment with human judgments under our validated schema.

**Answer-level performance.** Aggregate F1 ranking is ASEARCHER > Search-R1 > RESEARCH > Few-shot ≈ DEEPRESEARCHER > Base (Appendix E). Our primary analysis moves beyond outcomes to assess *process-level epistemic competencies*: evidence grounding (Section 4.2), recovery dynamics (Section 4.3), and calibration (Section 4.4). We identify agent-specific competencies that drive performance gains and expose the overestimation of RL training (Section 4.5).

## 4.2 EVALUATING JUSTIFIED REASONING VIA EVIDENCE GROUNDING

A fundamental criterion of agent competence is not merely producing the correct answer, but doing so through a reasoning process *explicitly grounded in retrieved evidence*. To measure this, we utilize the Reasoning Quality Index (RQI, defined in Section 3.3.1), which quantifies the proportion of an agent's reasoning steps that are supported by retrieved evidence.

**RL Training Fails to Develop Evidence-Grounded Reasoning.** Figure 3 (*Left*) presents the average RQI scores across models. Few-shot prompting achieves the highest reasoning quality (RQI = 0.27), outperforming all RL-trained agents. This reveals a *disconnect between answer-level success and reasoning groundedness*: RL training may optimize for correct final answers, but it fails to develop the epistemic reasoning skills to justify those answers with evidence-grounded reasoning.

**Plan Formation and State Assessment Are Core Reasoning Failures.** To understand *where* reasoning breaks down, we analyze performance by reasoning type (Figure 3, *Right*). Specifically,

- **Information Synthesis** emerges as a relative strength across models (e.g., ASEARCHER: 0.56), demonstrating agents' proficiency in summarizing and restating retrieved information.
- **Plan Formation** constitutes the most significant weakness for all agents (consistently scoring below 0.2), highlighting fundamental difficulties in breaking down complex queries and formulating coherent search strategies.
- **State Assessment** shows notable improvement in few-shot models (0.28), suggesting enhanced metacognitive capabilities compared to their RL-trained counterparts.

For detailed analysis of evidence-conditioned reasoning quality across different evidence states and reasoning types, see Appendix F.

## 4.3 RECOVERY ANALYSIS

This section evaluates whether models can effectively recover from low-quality evidence through adaptive search strategies.

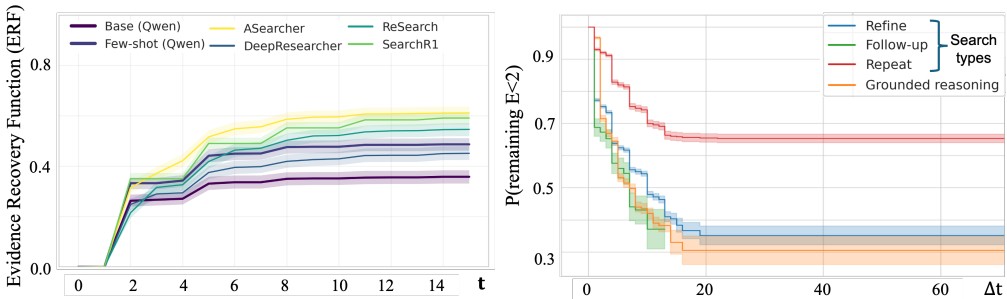

Figure 4: **Recovery Analysis.** *Left:* ERFs by model showing recovery from low to sufficient evidence (`E=2`) as turn $t$ increases. *Right:* Recovery efficiency by action type. Steeper curves indicate faster escape from low evidence states. REFINE and FOLLOW-UP enable fastest recovery, while REPEAT shows minimal improvement.

**Recovery Competence with ERF.** We first assess overall recovery competence using the Evidence Recovery Function (ERF, Equation (8)), which measures the cumulative probability of reaching sufficient evidence ($E = 2$) over time. As shown in Figure 4 (*Left*), ASEARCHER, which has the highest F1 score on answer correctness, shows superior recovery performance compared to other agents. In contrast, DEEPRESEARCHER, which has the lowest F1 among all RL-trained agents, shows the poorest recovery performance. This demonstrates that *effective algorithm design should prioritize developing adaptive evidence-seeking strategies so that agents can recover from insufficient evidence and improve final performance*.

**Refine and Follow-up Search Strategies Drive Effective Recovery.** To identify the *most effective search strategies* for recovery, we analyze how different action types affect recovery rates over time. We categorize all search and reasoning steps by their types. For each step $t$ of a specific type in trace, we measure the proportion of turns remaining low evidence states ($E < 2$) at subsequent turns ($t + \Delta t$). Given the variable-length traces and resulting right-censored data (traces ending before recovery occurs—when observation periods end before the outcome), we employ Kaplan-Meier survival analysis (Kaplan & Meier, 1958), which provides robust estimation of recovery probabilities despite incomplete observations. As shown in Figure 4 (*Right*), survival curves reveal that REFINE and FOLLOW-UP strategies enable the fastest recovery from low-quality evidence, while REPEAT provides minimal benefit. Additionally, GROUNDED REASONING also effectively improves evidence utilization in responses.

## 4.4 EVIDENCE-ALIGNED CALIBRATION

This section evaluates whether models calibrate their answering behavior to the ideal policy, where they answer when and only when it has good evidence, avoiding both overconfidence (answering with poor evidence) and overcautiousness (failing to answer despite good evidence).

**Evidence Quality Drives Answer Accuracy.** We first validate that evidence quality correlates with answer accuracy. As shown in Figure 5, RL-trained models achieve $31.6\%$ accuracy when answering with good evidence ($P(\text{correct}|\text{answer}, E = 2)$), compared to only $8.4\%$ accuracy when answering without supporting evidence. This significant difference demonstrates that *evidence quality is positively associated with answer correctness*.

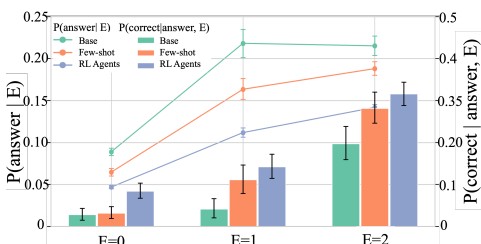

Figure 5: **Evidence State Drives Answer Probability and Accuracy.** *Lines:* Answer probability increases with evidence state. *Bars:* Answer accuracy improves with evidence state. RL-trained models show lower answering rate but higher accuracy with good evidence.

Interestingly, RL-trained models exhibit lower answering rates ($P(\text{answer}|E)$, formally defined in Equation 9) across all evidence states compared to base models. This suggests that RL training encourages models to be *more selective about when to provide final answers*, potentially reducing instances of overconfident responses.

Table 3: **Calibration Error Analysis.** traces categorized as: calibration error, overconfident, or overcautious. Lower values indicate better calibration. RL-trained agents show the lowest overconfident answer rate and lowest CE. **Bold** indicates best performance.

| Model | (1) Overconfident ↓ | (2) Overcautious ↓ | (3) Calibration Error ↓ |
|---|---|---|---|
| Base | 0.631 | 0.030 | 0.329 |
| Few-shot | 0.511 | **0.024** | 0.317 |
| RL-trained | **0.353** | 0.085 | **0.309** |

To further understand the calibration behavior, we measure calibration quality using **calibration error** (CE, Equation (10)) and analyze two specific failure modes to identify where models fail:

(1) **Overconfident answering**: providing a final answer when the trace never reached good evidence state ($E_{i,t} < 2$ for all $t$), indicating overconfidence;

(2) **Overcautious abstention**: failing to provide a final answer despite having reached good evidence state ($E_{i,t} = 2$), indicating underconfidence.

**RL Training Improves Calibration.** As summarized in Table 3, RL-trained models show substantial improvements in calibration behavior. They reduce overconfident answering from $63.1\%$ to $35.3\%$ and achieve the lowest overall calibration error (0.309). This demonstrates that RL training successfully teaches models to align their answering decisions with evidence quality, moving toward the ideal policy of answering only when evidence is sufficient. This finding contrasts with the earlier result that RL training degrades reasoning groundedness (Section 4.2), highlighting the *competency-specific nature* of RL training effects. For detailed analysis of individual RL-trained agents and evidence-conditioned answer timing patterns, see Appendix H.

## 4.5 EXPLOITING EPISTEMIC COMPETENCIES FOR PERFORMANCE GAINS

Our evaluation reveals distinct agent specializations: ASEARCHER excels in *evidence acquisition* and *recovery mechanisms* (highest overall F1 score), while SEARCH-R1 demonstrates superior **information synthesis** (RQI=0.63 for information synthesis) with **minimal overconfident answering** (see Section 4.4 and Appendix H). These differences are primarily driven by training objectives: SEARCH-R1 optimizes only for final answer correctness, achieving high accuracy (Jin et al., 2025) but sacrificing reasoning groundedness (Section 4.2), while ASEARCHER emphasizes data synthesis with failure and recovery strategies to teach adaptive re-planning (Gao et al., 2025), leading to superior evidence acquisition and recovery capabilities (Section 4.3). This motivates us to explore *agent synthesis*—using one agent's evidence collection as input for another's answer generation.

We provided agents with reasoning traces and evidence from others, then measured F1 score improvements. SEARCH-R1 emerges as the **most effective synthesizer** (+2.61 F1 on average), significantly outperforming other agents (see details in Appendix I). Surprisingly, Base achieved the highest F1 gains (+2.42 on average) when paired with other models for answer generation. This reveals that accuracy-only evaluation may **underestimate** Base's reasoning abilities while overstating the gains from RL training.

Our method reveals distinct agent profiles by systematically benchmarking their epistemic competencies, for example, SEARCH-R1's synthesis strength and conservative answering. These insights provide a reliable foundation for designing effective systems that capitalize on complementary agent strengths. Furthermore, we demonstrate that our epistemic competency framework can provide actionable inference-time feedback signals, achieving 8.4% F1 improvement on ASEARCHER-7B without training (see Appendix J). Overall, beyond outcome-based metrics, our approach delivers procedural evaluation that enables more interpretable assessments of agent competence.

## 5 CONCLUSION

**SeekBench** evaluates epistemic competence in LLM search agents through expert-annotated traces, revealing gaps in current evaluation approaches. Our evidence state framework and metrics (RQI, ERF, CE) uncovers agent-specific strengths masked by accuracy-only evaluation: Search-R1 excels at evidence synthesis, while Base models demonstrate stronger reasoning capabilities than accuracy metrics suggest. This work establishes epistemic competence as essential for developing reliable information-seeking agents. Future work should explore modular architectures combining complementary strengths and training approaches that improve reasoning alongside answer calibration.

**Acknowledgments.** This work is supported by Suzhou Frontier Science and Technology Program (Project SYG202310).

**Ethics Statement.** This research evaluates LLM search agents using publicly available datasets and involves only expert annotation of agent traces. Our framework aims to improve AI system reliability and transparency for developing more trustworthy information-seeking agents. The **SeekBench** dataset will be released with appropriate documentation while respecting licensing terms.

**Reproducibility Statement.** To ensure reproducibility of our findings, we provide comprehensive documentation of our methodology and evaluation framework. Our annotation schema and inter-annotator agreement analysis (Appendix A) establish the reliability of our epistemic competency measurements, with Cohen's $\kappa = 0.811$ for human annotators and strong LLM-judge alignment ($\kappa \geq 0.693$). The complete annotation guidelines and evaluation prompts are detailed in Appendix D, enabling replication of our step-level reasoning quality assessments. Our three core metrics—Reasoning Quality Index (RQI), Evidence Recovery Function (ERF), and Calibration Error (CE)—are formally defined with mathematical specifications in Section 3.3. The evaluation spans seven established QA benchmarks (NQ, TriviaQA, PopQA, HotpotQA, 2Wiki, MusiQue, Bamboogle) with 28,493 traces and 283,950 annotated steps across six agent variants (Qwen-2.5-7B-Instruct and few-shot, Search-R1, ReSearch, ASearcher, DeepResearcher). Data sanitization procedures to remove ambiguous questions and contamination cases are documented in Appendix B. All experimental results, including detailed agent-specific breakdowns and evidence-conditioned analyses, are provided in the main text and appendix. The **SeekBench** dataset of 190 expert-annotated traces with over 1,800 response steps will be made available as supplementary material, along with our annotation schema and evaluation code to enable community replication and extension of our epistemic competence framework.

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

CONTENTS

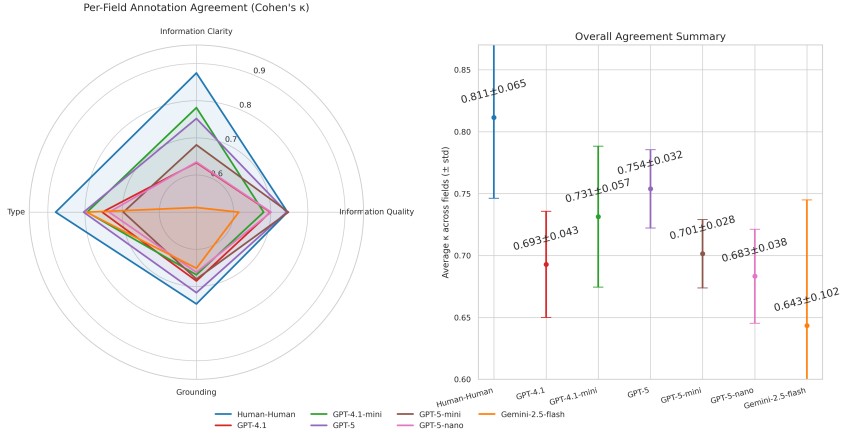

Figure 6: Inter-annotator agreement for **SeekBench**. (*Left*) Per-field annotation agreement across different competency dimensions. (*Right*) Average Cohen's $\kappa$ comparing human annotators with GPT-4.1, GPT-4.1-mini, and GPT-5, demonstrating strong alignment between expert human judgments and advanced LLM assessments.

## A    INTER-ANNOTATOR AGREEMENT ANALYSIS

**Annotation Schema Overview.** Our annotation schema (Figure 9) provides a structured framework for labeling agent traces, capturing both the functional role and epistemic quality of each step. The schema consists of two key dimensions: (1) **Functional Type**, which categorizes the cognitive purpose of reasoning and search steps (e.g., `StateAssessment`, `PlanFormation`, `InformationSynthesis` for reasoning; `InitialQuery`, `RefinedQuery`, `FollowUpQuery`, `RepeatQuery` for search), and (2) **Quality Attribute**, which evaluates epistemic soundness (e.g., `Groundedness` for reasoning, `Quality` and `Clarity` for search results, `Correctness` for final answers). This dual-dimensional structure enables comprehensive process-level evaluation by capturing both *what* agents are doing and *how well* they are doing it. The schema's reliability, as measured through inter-annotator agreement, establishes the foundation for our epistemic competence evaluation framework.

**Per-Field Agreement Analysis.** The left panel of Figure 6 demonstrates robust inter-annotator agreement across all four annotation fields. The **Functional Type** field achieves the highest agreement ($\kappa > 0.8$), indicating that annotators can reliably distinguish between different reasoning purposes (e.g., Information Synthesis vs. Plan Formation). The **Quality Attribute** field shows similarly strong agreement ($\kappa > 0.75$), confirming that evaluative judgments of epistemic soundness are consistently interpretable across annotators. These results establish that our schema captures meaningful, distinguishable patterns in agent reasoning behavior rather than subjective interpretations.

**Human-LLM Alignment Assessment.** The right panel reveals substantial alignment between human expert judgments and LLM assessments across all three evaluated models. Human annotators achieve the highest overall agreement ($\kappa = 0.811$), establishing the reference standard for annotation quality. Among LLM judges, GPT-5 demonstrates the strongest alignment with human experts ($\kappa = 0.754$), followed by GPT-4.1-mini ($\kappa = 0.731$) and GPT-4.1 ($\kappa = 0.693$). This progressive improvement across model versions suggests that more advanced language models can better approximate human reasoning patterns in epistemic evaluation tasks.

**Cost-Effectiveness Analysis for LLM Judges.** To evaluate the practical deployment of LLM-as-judge systems, we analyze the trade-off between annotation quality (measured by inter-annotator agreement, IAA) and per-trace cost (combining token cost and time cost) across six language models. Table 4 provides detailed per-trace cost and IAA metrics for all evaluated models. Figure 7 presents a Pareto frontier analysis, identifying models that achieve optimal balance between cost

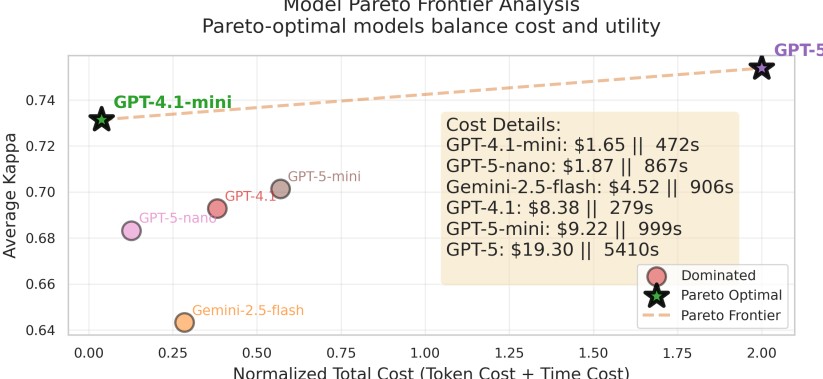

Figure 7: Model Pareto Frontier Analysis. Trade-off between normalized per-trace cost (x-axis) and average Cohen's $\kappa$ (IAA, y-axis) across language models. The x-axis is computed by min-max normalizing token and time costs per trace separately to $[0, 1]$, then summing with equal weight. GPT-4.1-mini and GPT-5 lie on the Pareto frontier, with GPT-4.1-mini offering optimal cost-effectiveness for large-scale deployment.

and utility. Our evaluation reveals that **GPT-4.1-mini** emerges as the **most cost-effective solution**, achieving strong human alignment (IAA $= 0.731 \pm 0.057$) at minimal per-trace cost (\$0.0087 and 2.48s), positioning it on the Pareto frontier alongside GPT-5. While GPT-5 achieves the highest agreement ($\kappa = 0.754 \pm 0.032$), its substantially **higher per-trace cost** (\$0.1016 and 28.47s) makes it less suitable for large-scale deployment. These results establish GPT-4.1-mini as the optimal choice for scalable evaluation frameworks that require both high-quality annotations and economic feasibility.

Table 4: Per-trace cost and Inter-Annotator Agreement (IAA) for LLM Judges on 190 sampled traces. Token cost (in USD\$ per trace), time cost (in seconds per trace), and IAA measured as Cohen's $\kappa$ with standard deviation. Models marked with † are Pareto-optimal.

| Model | Token Cost (\$/trace) | Time Cost (s/trace) | IAA ($\kappa \pm$ std) |
|---|---|---|---|
| GPT-4.1-mini[†] | 0.0087 | 2.48 | $0.731 \pm 0.057$ |
| GPT-5-nano | 0.0098 | 4.56 | $0.683 \pm 0.038$ |
| Gemini-2.5-flash | 0.0238 | 4.77 | $0.643 \pm 0.102$ |
| GPT-4.1 | 0.0441 | 1.47 | $0.693 \pm 0.043$ |
| GPT-5-mini | 0.0485 | 5.26 | $0.791 \pm 0.028$ |
| GPT-5[†] | 0.1016 | 28.47 | $0.754 \pm 0.032$ |

**Implications for Large-Scale Evaluation.** These agreement results establish the feasibility of deploying our annotation framework for comprehensive agent evaluation. The strong human-LLM alignment enables cost-effective scaling of our evaluation methodology, while the robust per-field agreement ensures that competency assessments reflect genuine behavioral differences rather than annotation artifacts. This validation is particularly crucial for our three core competencies (Groundedness, Recovery, and Calibration), as it confirms that these constructs can be reliably measured across diverse reasoning traces and evaluators.

## B  DATA SANITIZATION

To ensure the quality of our evaluation, we sanitize the test sets of our seven benchmark datasets using the following two criteria:

(1) **Ambiguous or Unanswerable Questions.** We discard questions where benchmark agents receive full credit for speculative answers, while a stronger reference model (GPT-4.1-mini) abstains with a justified explanation. For example, in response to the question *"Who developed the CPU?"*, a benchmark agent might confidently output *"John von Neumann"*, achieving EM=1. In contrast, GPT-4.1-mini responds: "Answer: I don't know; Reason: The information mentions figures like

`John von Neumann and J. Presper Eckert, but does not identify a single developer.`" These questions are excluded to avoid rewarding superficial matching over careful reasoning.

(2) **Data Contamination.** We discard questions where agents succeed (Pass@3) without issuing any search queries, as this indicates the question is likely part of the model's pre-training data. These are removed to focus evaluation on retrieval-dependent reasoning.

## C DATASET STATISTICS

### C.1 DATASET CONSTRUCTION

To validate our annotation schema and establish LLM-as-judge feasibility, we constructed an expert-annotated validation dataset of 190 traces. We sampled traces from the seven agents and seven QA benchmarks described in our experimental setup (Section 4.1). For each agent-dataset combination, we selected 2 correct and 2 incorrect answer traces, targeting an ideal sample size of $7 \times 7 \times 4 = 196$ traces. The final dataset contains 190 traces (6 fewer than ideal) because some traces, particularly from single-hop QA datasets, lack multiple search queries. We prioritize multi-query traces because our framework focuses on **process-level analysis** to evaluate how agents reason, adapt search strategies, and make decisions across multiple steps. This sampling strategy ensures our validation dataset captures the epistemic behaviors our framework is designed to assess.

### C.2 LARGE-SCALE EVALUATION DATASET

This section provides a comprehensive statistical analysis of the evaluation dataset, which comprises 28,493 traces across seven question-answering benchmarks. Figure 8 presents six complementary views of the dataset composition and annotation distributions, revealing key characteristics that inform our epistemic competency evaluation.

**Dataset Composition.** The evaluation dataset spans seven established question-answering benchmarks, providing diverse coverage of both single-hop and multi-hop reasoning tasks. As shown in Figure 8 (*top-left*), the dataset distribution is relatively balanced across sources: **PopQA** (28.9%) and **MusiQue** (26.3%) constitute the largest portions, followed by **HotpotQA** (12.5%) and **2Wiki-MultihopQA** (12.5%). **TriviaQA** (7.6%) and **NQ-Search** (7.6%) contribute smaller but substantial portions, while **Bamboogle** (4.6%) provides the smallest contribution. This distribution ensures that our evaluation framework is tested across diverse question types, from factual single-hop queries to complex multi-hop reasoning tasks requiring information synthesis across multiple sources.

**Information Quality and Clarity.** The quality and clarity of retrieved evidence are fundamental to evaluating epistemic competence, as they directly determine the evidence state $E_{i,t}$ used in our metrics. Figure 8 (*top-middle* and *top-right*) reveals critical challenges in the information landscape that agents must navigate:

- **Information Clarity:** A substantial majority of search results (62.5%, 48,134 instances) are categorized as **unclear**, meaning they contain ambiguous, vague, or confusing information that could match multiple entities or interpretations. Only 37.5% (28,905 instances) are classified as **clear**, indicating straightforward, unambiguous information. This distribution highlights the prevalence of ambiguous search results in information-seeking scenarios, emphasizing the importance of agents' ability to *handle uncertainty* and *adapt their search strategies*.
- **Information Quality:** The distribution of information sufficiency is nearly balanced, with 52.7% (40,559 instances) classified as **insufficient** and 47.3% (36,402 instances) as **sufficient**. This near-even split reflects the inherent difficulty of information-seeking tasks, where initial queries often fail to retrieve complete answers, requiring agents to demonstrate *recovery capabilities* through adaptive search refinement.

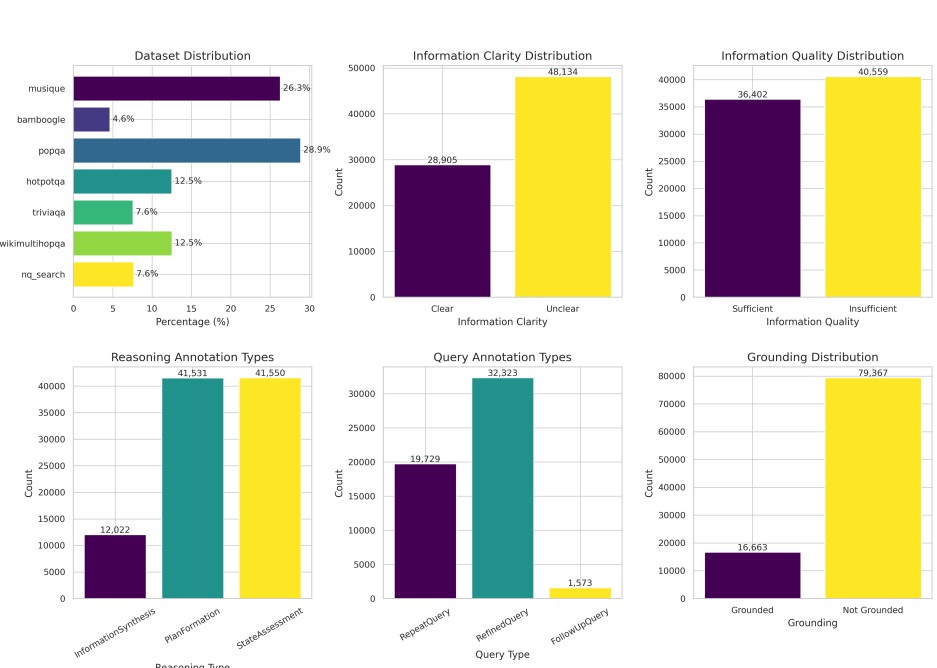

Figure 8: Comprehensive Dataset Statistics. (*Top-left:*) Distribution of traces across seven QA benchmarks, showing the relative contribution of each dataset. (*Top-middle:*) Information clarity distribution, categorizing search results as clear or unclear. (*Top-right:*) Information quality distribution, indicating whether retrieved evidence is sufficient or insufficient. (*Bottom-left:*) Distribution of reasoning annotation types across all reasoning steps. (*Bottom-middle:*) Distribution of query annotation types across all search steps. (*Bottom-right:*) Grounding distribution, showing the proportion of reasoning steps that are grounded versus not grounded in retrieved evidence.

**Reasoning Type Distribution.** Our annotation schema categorizes reasoning steps into three functional types that capture distinct cognitive processes in information-seeking. Figure 8 (*bottom-left*) shows that **Plan Formation** (41,531 instances, 43.7%) and **State Assessment** (41,550 instances, 43.7%) dominate the reasoning landscape, with nearly identical frequencies. These two types together account for 87.4% of all reasoning steps, indicating that agents spend substantial effort on metacognitive processes: identifying knowledge gaps (State Assessment) and formulating search strategies (Plan Formation). In contrast, **Information Synthesis** (12,022 instances, 12.6%) represents a smaller but critical component, where agents integrate retrieved evidence to form conclusions. This distribution reveals that effective information-seeking requires sophisticated planning and self-monitoring capabilities, not merely the ability to synthesize information once it is retrieved.

**Query Type Distribution.** Search behavior analysis reveals how agents adapt their information-seeking strategies. Figure 8 (*bottom-middle*) shows that **RefinedQuery** (32,323 instances, 60.3%) is the most common query type, demonstrating that agents frequently modify their search approach based on previous results. This adaptive behavior is essential for recovery from poor initial evidence. **RepeatQuery** (19,729 instances, 36.8%) represents a substantial portion of search behavior, indicating that agents sometimes persist with similar queries, which may reflect either strategic refinement or ineffective adaptation. **FollowUpQuery** (1,573 instances, 2.9%) is relatively rare, suggesting that agents infrequently employ exploratory follow-up questions that diverge from their primary search trajectory. This distribution underscores the importance of query refinement as a core mechanism for evidence recovery, while also highlighting the potential for improving agents' exploratory search capabilities.

**Grounding Distribution.** The grounding analysis evaluates whether reasoning steps are supported by retrieved evidence, which is central to the **Groundedness** competency. Figure 8 (*bottom-right*) reveals a critical finding: only 17.3% (16,663 instances) of reasoning steps are classified as **grounded**, while 82.7% (79,367 instances) are **not grounded**. This substantial imbalance indicates that the majority of agent reasoning steps lack direct support from retrieved evidence, representing a fundamental epistemic challenge. This finding aligns with our RQI analysis (Section 4.2). The prevalence of ungrounded reasoning highlights the critical need for evaluation frameworks that assess **epistemic competence** beyond answer-level accuracy, as agents may produce correct answers through epistemically unsound reasoning processes.

**Implications for Epistemic Competence Evaluation.** These statistical patterns reveal several key insights that inform our evaluation framework. First, the high proportion of unclear and insufficient evidence (62.5% unclear, 52.7% insufficient) establishes a challenging information landscape where agents must demonstrate *robust recovery capabilities*. Second, the dominance of `PlanFormation` and `StateAssessment` reasoning types (87.4% combined) suggests that metacognitive capabilities are central to effective information-seeking, yet our RQI analysis reveals these are precisely the reasoning types where agents struggle most. Third, the overwhelming prevalence of ungrounded reasoning (82.7%) confirms that evidence-grounded reasoning is a critical competency gap that current agents fail to address, validating the necessity of process-level evaluation frameworks like **Seek-Bench**.

## D  LLM-AS-JUDGE FOR SEEKBENCH

This section presents the comprehensive LLM-as-judge used in **SeekBench** to evaluate agent reasoning, search behavior, and answer quality. The schema is organized into: reasoning types with grounding evaluation, search behavior, search result quality.

### D.1  REASONING TYPE ANNOTATION AND GROUNDING EVALUATION

The reasoning annotation schema categorizes agent reasoning steps into four functional types and evaluates their grounding. This comprehensive evaluation helps identify both the cognitive function of reasoning steps and whether they are properly supported by evidence.

---

**Reasoning Type Classification and Grouding Evaluation**

You are an expert cognitive scientist and evidence-based critical thinking expert. Your task is to classify the reasoning type of an agent's step and evaluate its grounding based *only* on the evidence it had at the time.
Context: The Agent's Goal (Original Question):

`{question}`

Evidence: The Search Results the Agent Had Access To:

`{search_evidence_json}`

Agent's Reasoning Text to Analyze:

`"{reasoning_text}"`

---

Task:
   1. Classify the reasoning type:
      - **StateAssessment**: Assess the current knowledge state, usually identifying a knowledge gap.
      - **PlanFormation**: The agent is forming a plan of action.
      - **InformationSynthesis**: Synthesize new information (from search results) to form a conclusion.
   2. Evaluate grounding:
      - Extract the atomic factual premises from the step (skip meta/plan-only wording that contains no factual claim).

---

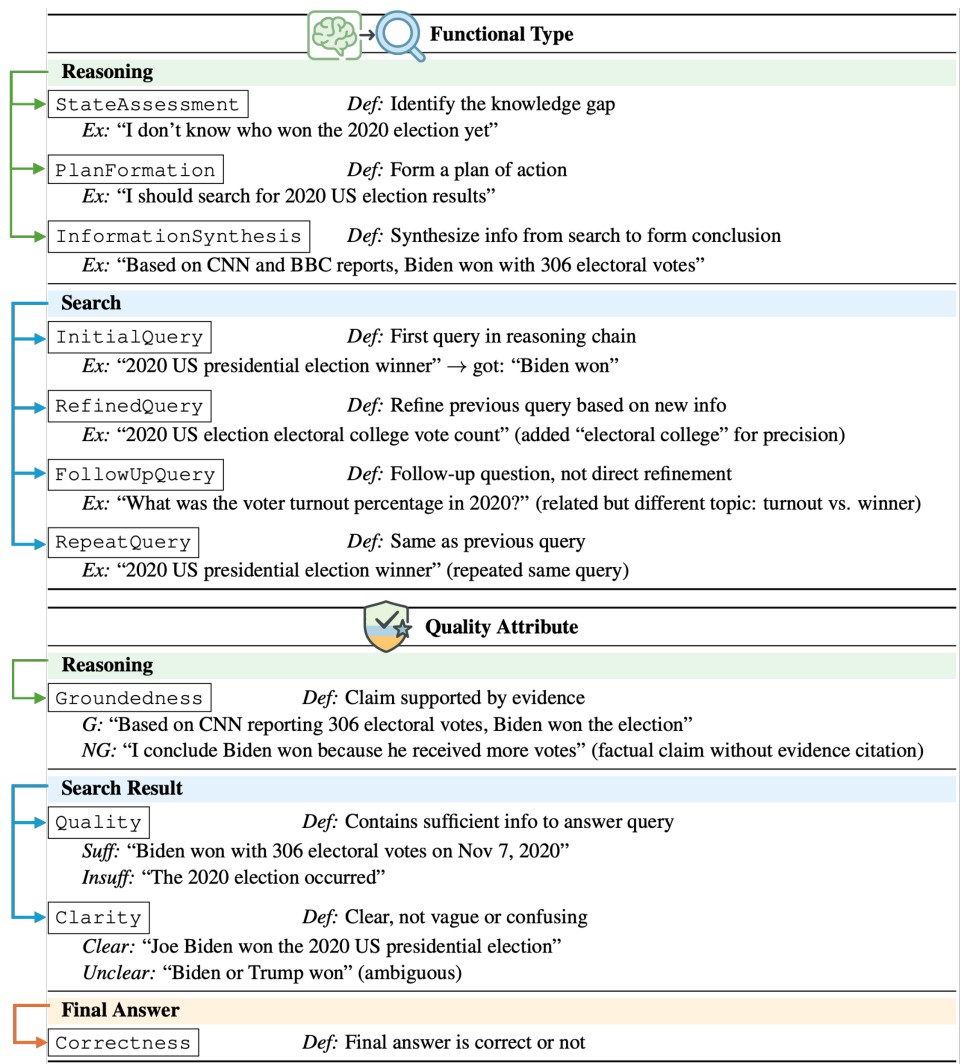

Figure 9: Annotation schema overview with definitions and examples. *Abbreviations:* Def=Definition, Ex=Example, G=Grounded, NG=Not Grounded, Suff=Sufficient, Insuff=Insufficient.

> - For each premise, find a direct supporting span in the provided evidence. If no exact or near-verbatim support exists, mark that premise as unmatched.
> - Decide the label with STRICT rules:
>   - **Grounded**: ALL atomic premises are supported by explicit evidence spans.
>   - **Not Grounded**: ANY atomic premise lacks a supporting span; OR the step contains only meta/plan text without factual premises.
>
> Additional rules for grounding:
> - QUESTION anchor alone is NOT sufficient for Grounded; do not label as grounded solely for restating the task/intent.
> - Superlatives/temporal/quantitative claims (e.g., last/first/only, years, counts) require explicit evidence spans.
>
> ---
>
> Your Final Output:
>
> ```
> {
>   "reasoning_type": "StateAssessment",
>   "grounding": "Grounded",
>   "anchor_type": "EVIDENCE",
> ```

```
  "justification": "brief explanation"
}
```

## D.2 SEARCH BEHAVIOR ANNOTATION

The search annotation schema categorizes agent search queries into four behavioral types:

---

**Search Behavior Classification**

You are an expert information retrieval specialist. Your task is to classify the type of search query issued by the agent.
Current Search Query:

```
{current_query}
```

Previous Search Query (if any):

```
{previous_query}
```

---

Task: Classify the search query type:
- **InitialQuery**: The agent is issuing its first query in a reasoning chain.
- **RefinedQuery**: The agent is refining a previous query based on new information.
- **FollowUpQuery**: The agent is asking a follow-up question that is not a direct refinement.
- **RepeatQuery**: A query that is the same as the previous query.

---

Your Final Output:

```
{
  "search_type": "InitialQuery",
  "justification": "brief explanation"
}
```

---

## D.3 SEARCH RESULT QUALITY ASSESSMENT

This prompt evaluates the quality and clarity of search results retrieved by the agent. It helps identify when agents work with insufficient or ambiguous information.

---

**Search Result Analysis Prompt**

You are an expert data analyst. Your task is to evaluate the quality of a search result based on the query that produced it.
Search Query:

```
{query}
```

Search Result Documents:

```
{documents_json}
```

---

Your Task: Analyze the search result's sufficiency and clarity.
1. Information Quality: Does the result contain enough information to likely answer the user's implicit question in the query? Choose one:
- Sufficient: The answer seems to be present.
- Insufficient: The answer is likely not here.
2. Information Clarity: Is the information clear or does it create confusion? Choose one:
- Clear: The information is straightforward and addresses one subject.
- Unclear: The results mention multiple distinct entities that could match the query (e.g., two movies with the same title) or the information is vague.

---

> Your Final Output: Your response must be a single, valid JSON object with the following attributes:
> - `information_quality`: Either "Sufficient" or "Insufficient"
> - `information_clarity`: Either "Clear" or "Unclear"
> - `clarity_justification`: Brief explanation for your clarity rating

# E    ACCURACY-LEVEL PERFORMANCE

| Agent | Overall F1 (%) |
|---|---|
| ASearcher | 39.77 |
| Search-R1 | 39.29 |
| ReSearch | 38.30 |
| Few-shot | 36.04 |
| DeepResearcher | 36.00 |
| Base | 33.50 |
| CoT | 32.72 |
| ReAct | 31.25 |

Table 5: Overall F1 performance across agent variants. Trained agents consistently outperform the base model, with ASearcher achieving the highest score.

Table 5 reports the aggregate F1 scores across all evaluated agents. We observe that all trained agents outperform the base Qwen model, with ASearcher achieving the best performance (39.8%). Search-R1 and ReSearch follow closely, while Few-shot prompting and DeepResearcher attain comparable scores.

# F    EVIDENCE-GROUNDED REASONING ANALYSIS

This section provides detailed analysis of evidence-grounded reasoning capabilities for Section 4.2, examining how agents ground their reasoning in retrieved evidence and identifying critical gaps in epistemic alignment. We present two complementary analyses: (1) evidence-conditioned reasoning quality across different evidence states, and (2) type-specific reasoning capabilities that reveal heterogeneous grounding patterns across reasoning skills.

## F.1    EVIDENCE-ALIGNED REASONING: DO AGENTS GROUND THEIR INFERENCE IN WHAT THEY KNOW?

To evaluate whether agents ground their reasoning in retrieved evidence, we analyze the expected groundedness of reasoning steps conditioned on the agent's **evidence state** $E \in \{0, 1, 2\}$. The quantity $\mathbb{E}[G_{i,t} \mid E_{i,t} = k]$ measures how reliably an agent produces well-supported reasoning at each evidence level $k$. An **epistemically sound** agent should avoid unsupported reasoning when $E = 0$, provide partial grounding at $E = 1$, and fully leverage complete evidence when $E = 2$. This conditional analysis enables assessment of **epistemic alignment**: whether agents reason more confidently only when they possess sufficient evidence.

**Base models show better epistemic alignment than specialized agents.** Empirical results in Figure 10 reveal substantial variation across models. Most agents demonstrate appropriate behavior at $E = 0$, with $\mathbb{E}[G \mid E = 0] \approx 0.07$–$0.09$, indicating minimal hallucinated reasoning. However, SEARCH-R1 exhibits significant **epistemic misalignment**, with elevated groundedness even under insufficient evidence ($\approx 0.10$ at $E = 1$ and $0.14$ at $E = 2$), suggesting grounded reasoning. In contrast, BASE and FEW-SHOT variants demonstrate the clearest **evidence-conditioned reasoning**, with groundedness rising from 0.49 ($E = 1$) to 0.64 ($E = 2$), indicating effective epistemic modulation. ASEARCHER also shows notable improvement ($0.50 \rightarrow 0.55$), while RESEARCH and DEEPRESEARCHER stagnate around 0.47–0.51, failing to capitalize on stronger evidence. These

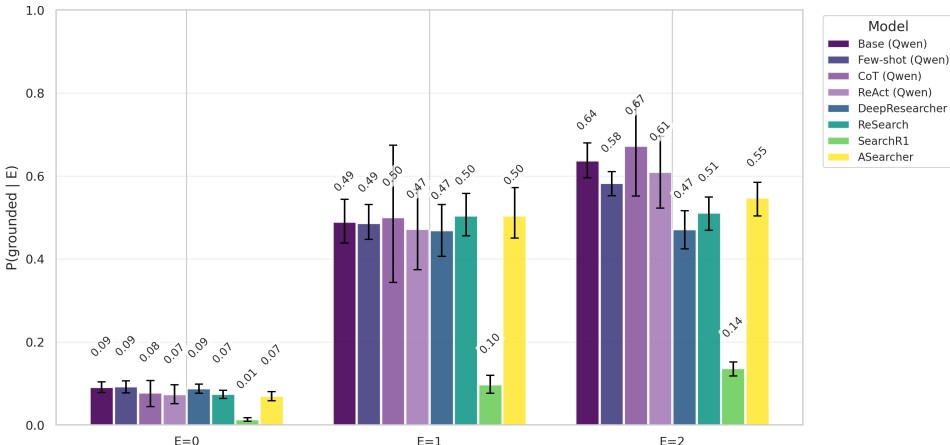

Figure 10: Evidence-conditioned reasoning quality for evidence state $k \in 0, 1, 2$ across search agents. Bars denote 95% confidence intervals. The quantity reflects the expected groundedness of reasoning steps given the epistemic evidence state $E$. Higher values at $E = 2$ indicate effective evidence utilization.

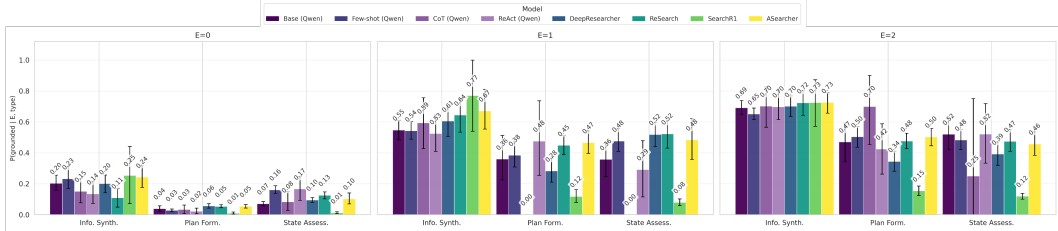

Figure 11: **Type-Level Evidence-Conditioned Groundedness.** Expected groundedness for each reasoning type $\tau$ (Information Synthesis, Plan Formation, State Assessment) under evidence levels $E = 0, 1, 2$. Bars show 95% confidence intervals. Models exhibit heterogeneous capabilities in grounding specific reasoning skills in available evidence.

results demonstrate the necessity for evaluation metrics like **RQI** that isolate whether reasoning reflects the agent's actual knowledge state.

**Case Study: Correct Answer with Ungrounded Reasoning.** We examine a case where a ReSearch agent correctly answers "Who won the first celebrity big brother on channel 5?" despite completely ungrounded reasoning. After retrieved the first evidence, the agent retrieves conclusive evidence stating "Celebrity Big Brother 1... concluded on 16 March 2001 when comedian Jack Dee was crowned the winner." Despite having the answer, the agent *ignores this evidence and conducts unnecessary searches*, stating: "I need to clarify which Big Brother series I am referring to... Now, I have to find out the winner of that show." The agent eventually answers "Jack Dee" correctly, but through an epistemically unsound process. This demonstrates why accuracy metrics alone fail to capture critical reasoning deficiencies.

### F.2  Type-Specific Evidence Alignment: Which Reasoning Skills Are Evidence-Grounded?

We further decompose agent reasoning groundedness by reasoning type $\tau \in \{\texttt{IS}, \texttt{PF}, \texttt{SA}\}$, leveraging the *Type-Level Reasoning Quality Index* from Definition 3.3 and the evidence-state decomposition in Equation (6).

Figure 11 illustrates the groundedness of each reasoning type across three evidence states. Our analysis reveals several significant patterns:

First, **Information Synthesis (IS)** demonstrates the strongest evidence-responsiveness across all models. With complete evidence ($E$=2), IS steps achieve superior groundedness, indicating robust

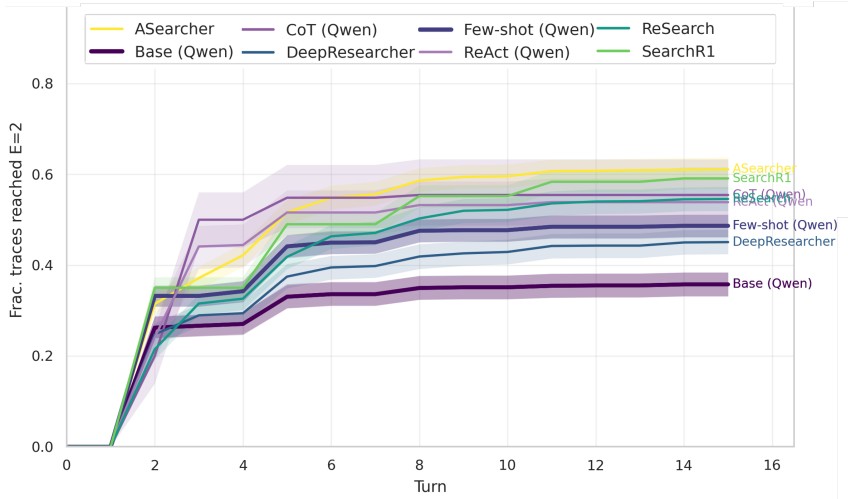

Figure 12: **Comprehensive Recovery Analysis.** Evidence Recovery Function (ERF) showing the cumulative fraction of traces reaching sufficient evidence state ($E = 2$) across all evaluated agents, including RL-trained agents (ASearcher, Search-R1, ReSearch, DeepResearcher), base models (Base), and prompting strategies (Few-shot, CoT, ReAct). Steeper curves indicate faster recovery from low-quality evidence.

capabilities in aggregating retrieved information. Even with partial evidence ($E$=1), agents maintain moderate IS groundedness, suggesting effective utilization of incomplete knowledge.

In contrast, **Plan Formation (PF)** and **State Assessment (SA)** exhibit substantially lower *groundedness* even with complete evidence. For PF, only ASEARCHER and RESEARCH exceed 0.5 at $E$=2, while others (e.g., SEARCH-R1, DEEPRESEARCHER) remain below 0.4, revealing *fragile decision-making processes* despite available knowledge. Similarly, **SA** demonstrates critical limitations: although scores improve under $E$=2, most models *underperform* relative to IS, with several agents (e.g., SEARCH-R1, DEEPRESEARCHER) showing *minimal evidence-responsiveness* between $E$=1 and $E$=2. Notably, SEARCH-R1 performs comparatively well for IS across all evidence levels but demonstrates *exceptionally poor grounding* for PF and SA (0.15 and 0.12 at $E$=2, respectively), suggesting *specialized evidence synthesis* but *narrowly constrained reasoning capabilities*.

These findings demonstrate that only specific reasoning capabilities (particularly synthesis) are consistently *grounded* in retrieved information. This underscores the necessity for developing *evidence-grounded reasoning policies*, especially for higher-order cognitive functions like plan formation and state assessment that currently show significant *epistemic disconnection* from retrieved knowledge.

## G    RECOVERY ANALYSIS

This section provides detailed analysis of evidence recovery capabilities, extending the main text analysis (Section 4.3) with additional agent comparisons including Chain-of-Thought (CoT) and ReAct prompting strategies.

**Extended Agent Comparison.** Figure 12 presents the Evidence Recovery Function (ERF) for all evaluated agents, including the additional CoT and ReAct prompting strategies. The results reveal a clear performance hierarchy: SEARCH-R1 demonstrates the best recovery performance, followed by **CoT**, RESEARCH, **ReAct**, FEW-SHOT, DEEPRESEARCHER, and BASE (in descending order). Notably, **CoT** outperforms several RL-trained agents (RESEARCH, DEEPRESEARCHER) and base models, achieving the second-highest recovery rate. **ReAct** also shows competitive recovery performance, outperforming FEW-SHOT, DEEPRESEARCHER, and BASE. This finding reveals that while RL-trained agents like ASEARCHER and SEARCH-R1 achieve superior recovery through explicit search adaptation mechanisms, prompting strategies like **CoT** can also demonstrate effective evidence recovery capabilities, potentially through their structured reasoning approach that enables better query refinement.

## H  EVIDENCE CALIBRATION ANALYSIS

To further understand the epistemic calibration capacity of RL-trained agents, we analyze four RL-based agents: ASEARCHER, SEARCH-R1, RESEARCH, and DEEPRESEARCHER. Each model exhibits distinct behavior patterns in evidence-grounded answering.

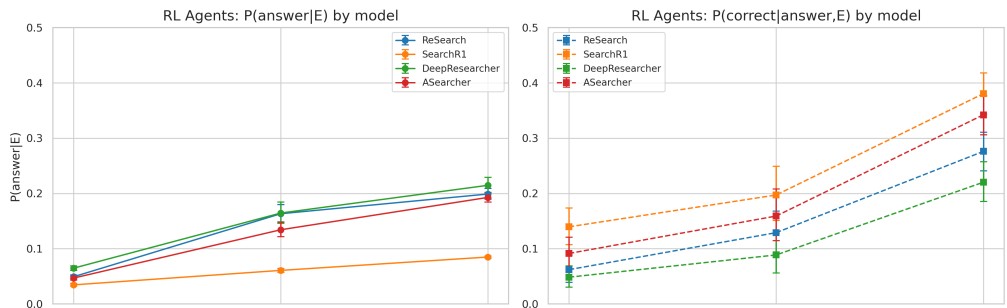

Figure 13: **Evidence-Calibrated Answering by RL Agents.** *Left:* Answering propensity for each RL agent under different evidence levels (E=0/1/2). *Right:* Answering accuracy for each RL agent under different evidence levels (E=0/1/2).

As shown in Figure 13 (*Left*), all agents show increased answering with stronger evidence, suggesting basic epistemic alignment, but they vary in *evidence gradient*—the increase from E=0 to E=2. ASEARCHER and DEEPRESEARCHER exhibit higher gradients, indicating *stronger sensitivity to epistemic evidence*. However, all agents maintain relatively low absolute response rates even with sufficient evidence. ASEARCHER and DEEPRESEARCHER reach 19–21%, while SEARCH-R1 remains at 8.5%, suggesting more conservative behavior.

**Calibration vs Accuracy** As noted in Section E, ASEARCHER achieves the highest overall F1 score, followed by SEARCH-R1, RESEARCH, and DEEPRESEARCHER.

To evaluates whether agents can defer or respond based on the epistemic adequacy of observed evidence. As shown in Figure 13, across trained agents, we observe distinct calibration profiles:

- ASEARCHER demonstrates superior evidence sensitivity coupled with high accuracy. It responds predominantly when sufficient evidence is available ($E=2$) at a substantial rate (19.3%) while minimizing overconfident responses. As shown in Figure 13 (*Right*), it achieves the second highest conditional accuracy $P(\text{correct}|\text{answer}, E = 2)$. This strategic alignment between evidence-based answer timing and correctness yields optimal performance.
- SEARCH-R1 achieves the highest accuracy when answering with complete evidence (Figure 13, *Right*), but exhibits extreme conservatism, answering in merely 8.5% of high-evidence states (Figure 13, *Left*). While this demonstrates exceptional calibration awareness, the excessive caution significantly constrains overall performance, representing a clear trade-off between coverage and precision.

We conclude that well-calibrated agent behavior requires satisfying two critical conditions: (1) deferring responses until evidence strength reaches sufficient levels (e.g., $E=2$), and (2) producing correct answers when responding, demonstrating effective utilization of the available evidence. This dual requirement highlights the challenge of balancing epistemic caution with informational utility.

**RL Training Reduces Calibration Errors.** Table 6 provides a detailed breakdown of calibration performance across individual RL-trained agents. While all RL agents show substantial improvements over base models in reducing overconfident answering, ASEARCHER achieves the lowest overall calibration error (0.302), closely followed by DEEPRESEARCHER and RESEARCH (both 0.305). Notably, SEARCH-R1 exhibits the most conservative behavior with the *lowest* overconfident answeringrate (0.226) but *highest* overcautious rate (0.187), suggesting a trade-off between different types of calibration failures. These results demonstrate that RL training consistently improves evidence-based decision making, though specific training approaches yield different calibration profiles.

Table 6: **Calibration Error Breakdown by Agent Type.** Trajectories categorized as: overconfident answering (answering before sufficient evidence), overcautious abstention (failing to answer despite strong evidence), and overall calibration error. Lower values indicate better calibration. ASearcher show the lowest calibration errors. **Bold** indicates best performance in each category.

| Model | (1) Overconfident ↓ | (2) Overcautious ↓ | (3) Calibration Error ↓ |
|---|---|---|---|
| Base | 0.631 | 0.030 | 0.329 |
| Few-shot | 0.511 | 0.024 | 0.317 |
| CoT | 0.731 | **0.006** | 0.351 |
| ReAct | 0.660 | 0.012 | 0.336 |
| ASearcher | 0.343 | 0.044 | **0.302** |
| DeepResearcher | 0.461 | 0.048 | 0.309 |
| ReSearch | 0.406 | 0.047 | 0.305 |
| Search-R1 | **0.226** | 0.187 | 0.319 |

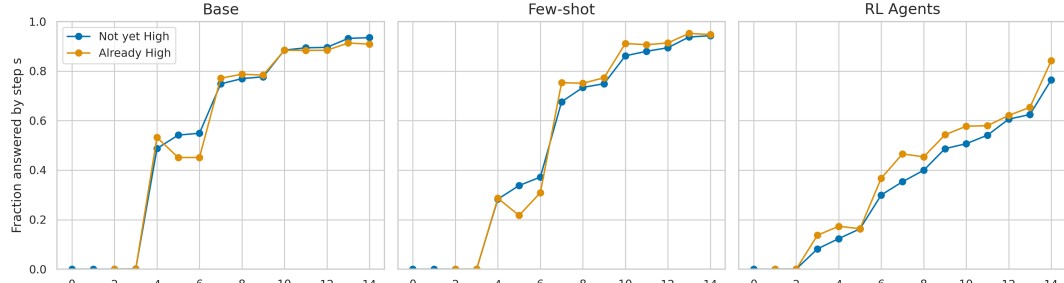

Figure 14: **Evidence-Conditioned Answer Timing Analysis.** For each model group, we plot the fraction of answered trajectories over time (x-axis: turn number), split by whether high evidence ($E=2$) has been observed. If agents defer until sufficient evidence, the orange curve (already high) should rise earlier than the blue curve (not yet high). However, all models show little separation between the two, confirming widespread *overconfident answering behavior*.

**Do agents defer answering until strong evidence arrives?** To evaluate whether agents appropriately delay answering until they have observed sufficient evidence, we analyze the timing of answers across different evidence states. Figure 14 presents a temporal analysis comparing the cumulative fraction of answers over time for two distinct trajectory groups: those where agents have already encountered strong evidence ($E=2$) versus those where they have not.

In an *ideally calibrated system*, we would expect agents to predominantly answer after observing strong evidence, resulting in a clear separation between trajectories—specifically, a higher orange curve (evidence already observed) and a lower blue curve (evidence not yet observed). However, our analysis reveals that BASE and FEW-SHOT models demonstrate *minimal separation* between these curves—indicating that answers are generated with similar timing regardless of evidence availability. RL-trained agents, while showing marginal improvement, still exihibits overconfident answering in 76.5% of trajectories before reaching sufficient evidence ($E=2$).

This finding highlights a *fundamental calibration deficiency*: current models consistently make overconfident decisions without aligning their answer timing with *epistemic sufficiency*.

## I  AGENT SYNTHESIS: LEVERAGING EPISTEMIC COMPETENCIES FOR ANSWER GENERATION

Our comprehensive evaluation reveals several critical insights into agent capabilities: ASEARCHER demonstrates superior performance in **evidence acquisition** and **recovery mechanisms** (achieving the highest overall F1 score), while SEARCH-R1 exhibits exceptional proficiency in **information synthesis** (with a RQI score of 0.63) coupled with **minimal overconfident answering behavior** (as discussed in Section 4.4). This observed specialization of epistemic competencies motivated

| | Policy (P) | | | | | | | | Overall |
|---|---|---|---|---|---|---|---|---|---|
| Synth. (S) | Search-R1 | ASearcher | DeepRes. | ReSearch | Base | Fewshot | CoT | ReAct | Avg. $\Delta$F1 |
| Search-R1 | – | −0.08 | +0.19 | +1.66 | +3.50 | +1.10 | +6.0 | +5.9 | **+2.61** |
| ASearcher | +0.38 | – | −0.73 | −0.39 | +0.93 | +1.46 | +5.9 | +5.4 | +1.85 |
| DeepRes. | −0.63 | +0.20 | – | +0.00 | +2.24 | +1.45 | +6.3 | +5.9 | +2.21 |
| ReSearch | −0.42 | +0.13 | +0.36 | – | +2.99 | +1.89 | +5.5 | +4.9 | +2.19 |

Table 7: **Agent Synthesis Performance.** Each cell shows F1 score improvement ($\Delta$F1) when using row agent (S) as synthesizer to generate answers based on evidence collected by column agent (P). Positive values indicate the synthesizer improved upon the original policy's performance. Search-R1 demonstrates the highest overall improvement (+2.61 F1) across all evidence sources.

us to explore the potential of *agent synthesis* where we leverage one agent's evidence collection capabilities as input for another agent's answer generation process.

**Synthesizer Evaluation Methodology.** To test this hypothesis, we evaluate each agent as a *synthesizer* by providing it with the complete reasoning traces, search results, and evidence from other agents' trajectories. The synthesizer's task is to generate a final answer based solely on this information, without performing additional searches. This setup isolates the agent's ability to synthesize information from existing evidence, separate from its search and retrieval capabilities.

**Search-R1 Emerges as the Superior Synthesizer.** As shown in Table 7, **Search-R1** delivers the largest average F1 gain (+2.61), significantly outperforming other agents such as ASEARCHER (+1.85), DEEPRESEARCHER (+2.21), and RESEARCH (+2.19). This result aligns with our earlier findings that Search-R1 exhibits strong information synthesis capabilities (RQI = 0.63 for Information Synthesis steps) and conservative answering behavior (lowest overconfident answering rate), and it persists even when synthesizing CoT or ReAct evidence traces.

The superior synthesis performance of Search-R1 can be attributed to its **specialized reasoning capabilities**. Despite its low overall RQI (0.08), Search-R1 demonstrates particular strength in **information synthesis** when provided with clear evidence. Its **conservative answering behavior**, while limiting coverage in standalone scenarios, becomes an advantage in synthesis tasks where it can carefully evaluate and integrate information from multiple sources before providing a final answer.

**Hidden Behaviors: How Accuracy-Level Evaluation Obscures Profound Reasoning Capabilities.** Surprisingly, several agents demonstrate stronger capabilities as evidence sources than their standalone F1 scores suggest. BASE evidence collection achieved substantial F1 gains when paired with other models for answer generation (up to +3.50 F1 improvement with Search-R1), despite having the lowest standalone F1 score (33.5%). Similarly, **CoT** and **ReAct**, which show lower standalone performance (32.72% and 31.25% F1, respectively), enable even larger synthesis improvements (up to +6.3 F1 with CoT and +5.9 F1 with ReAct). This reveals that final accuracy metrics can be misleading when used in isolation, as they obscure critical process-level competencies. These agents may be collecting high-quality evidence or producing well-structured reasoning traces but **struggling with final answer synthesis**, a nuance completely missed by traditional evaluation methods that focus solely on final answer accuracy rather than decomposing the reasoning process into its constituent competencies. The agent synthesis framework enables us to identify these hidden strengths and leverage complementary capabilities across different agent architectures.

**Implications for Agent Design.** These findings demonstrate that our benchmark and evaluation framework enables *modularization of agent-specific epistemic competencies* to create more effective information-seeking systems. This represents a significant advance for process-level evaluation of agents compared with traditional answer-level evaluation, enabling the identification and combination of complementary strengths across different agent architectures.

## J  INFERENCE-TIME FEEDBACK USING EPISTEMIC COMPETENCIES

In this section, we leverage the LLM-as-judge framework to provide inference-time feedback signals to improve agent behavior.

**Inference-Time Feedback Signals.** We augment ASEARCHER-7B with *inference-time feedback signals* derived from our epistemic competency framework. Specifically, we incorporate three process-level supervision signals computed at each turn: (1) **groundedness** feedback indicating whether reasoning steps are supported by retrieved evidence, (2) **evidence state** signals ($E_{i,t} \in \{0, 1, 2\}$) encoding the sufficiency and clarity of retrieved evidence, and (3) **calibration** feedback on whether the agent's answering behavior aligns with evidence quality. These signals are computed using our validated LLM-as-judge framework and provided as real-time guidance during inference, enabling the agent to adjust its behavior based on process-level epistemic feedback without training.

**Results and Implications.** Our experiments reveal that incorporating epistemic feedback signals at inference time significantly improves agent performance. The augmented ASEARCHER-7B achieves a **8.4% increase in final F1 score** compared to the baseline without feedback signals. More importantly, we observe substantial improvements across all three epistemic competencies: (1) **groundedness** (RQI) increases by 13.3%, indicating better evidence-supported reasoning, (2) **recovery** (ERF) improves by 6.5%, demonstrating more effective adaptation to insufficient evidence, and (3) **calibration** (CE) decreases by 5.8%, showing better alignment between answering behavior and evidence quality.

## K  EVALUATION ON GAIA

In this section, we evaluate the epistemic competencies of ASearcher and WebSailor on GAIA, and GPT-5-mini on GAIA with Web search tools.

We have evaluated WebSailor (Li et al., 2025a) and ASearcher (with web browsing + visit capabilities) on GAIA [1] with Pass@2 scores across different model sizes[2]. Our framework's tool selection follows epistemic scope rather than tool diversity. We include **web browsing and visit capabilities** (as in WebSailor) because these require epistemic evaluation—the quality of retrieved information depends on reasoning processes and evidence interpretation, not objective correctness.

**Answer-level Performance.** At 7B scale, ASEARCHER achieves 16.5% while WEBSAILOR achieves 18.2%. At 32B scale, ASEARCHER achieves 29.2% while WEBSAILOR achieves 32.4%. In addition, we evaluate GPT-5-mini on GAIA, achieving 15.7% Pass@2.

**Epistemic Metrics Analysis** We apply our epistemic competency framework to analyze ASEARCHER and WEBSAILOR behavior across the three metrics. Table 8 presents the epistemic metrics. For 32B scale, ASEARCHER demonstrates superior **groundedness** (RQI = 0.28) and **recovery** (ERF = 58% by Turn 8) compared to WEBSAILOR. However, WEBSAILOR shows better **calibration** (CE = 0.31) than ASEARCHER (CE = 0.35), indicating more conservative answering behavior. These metrics reveal distinct epistemic competency profiles that explain the performance differences: ASEARCHER's strength in evidence acquisition and recovery compensates.

We then apply our epistemic competency framework to analyze GPT-5-mini's behavior across the three metrics: **groundedness** (RQI), **recovery** (ERF), and **calibration** (CE). Based on this analysis, we provide inference-time feedback signals (as described in Appendix J), which improves the final score to 22.5%, a 43% relative improvement.

---

[1]https://huggingface.co/datasets/gaia-benchmark/GAIA
[2]Note that WebSailor does not have an open-sourced 14B variant, so we evaluate the available model sizes

Table 8: Epistemic Metrics for ASearcher and WebSailor on GAIA (7B and 32B) and GPT-5-mini on GAIA with Web search tools. RQI (groundedness), ERF recovery rate by Turn 8, and CE (calibration error, lower is better).

| Model | 7B | | | 32B | | | GPT-5-mini | | |
|---|---|---|---|---|---|---|---|---|---|
| | RQI | ERF | CE | RQI | ERF | CE | RQI | ERF | CE |
| ASEARCHER | 0.22 | **42%** | 0.38 | 0.28 | **58%** | 0.35 | – | – | – |
| WEBSAILOR | 0.15 | 35% | **0.33** | 0.19 | 45% | **0.31** | – | – | – |
| GPT-5-mini | – | – | – | – | – | – | **0.45** | 42% | 0.34 |

## L    DISCUSSION: UNDERSTANDING METRIC TRADE-OFFS

Our evaluation framework reveals distinct trade-offs between epistemic competencies that traditional accuracy-only evaluation fails to capture. These insights are critical for both interpreting agent performance and designing effective systems.

**The Accuracy-Reasoning Trade-off.** We observe a particularly concerning inverse relationship between answer accuracy (F1) and reasoning quality (RQI) among RL-trained agents. While RL training improves final answer correctness and calibration, it simultaneously degrades evidence-grounded reasoning quality. This reveals a fundamental tension: *agents can be optimized to produce correct answers without developing sound reasoning capabilities*—a critical consideration for AI safety and interpretability.

**Calibration vs. Reasoning Quality.** RL-trained models demonstrate better calibration (lower CE) despite worse reasoning groundedness (lower RQI), indicating that *well-calibrated agents may still produce poorly justified reasoning*. This highlights the necessity of evaluating both when to answer (calibration) and how to reason (groundedness) as separate competencies.

**Implications for Agent Selection and Design.** These trade-offs directly impact deployment decisions: applications requiring high accuracy may favor ASEARCHER despite reasoning limitations; those requiring interpretable reasoning may prefer Few-shot models despite lower accuracy; and applications demanding both may benefit from agent synthesis approaches (Section 4.5). Future agent development should explicitly address these trade-offs through multi-objective optimization and potentially modular architectures that separate evidence acquisition, reasoning, and decision-making components rather than optimizing solely for accuracy. Furthermore, our epistemic competency metrics (RQI, ERF, CE) can drive automatic orchestrators that monitor agent behavior in real-time and dynamically switch to the best-suited agent for each stage of the information-seeking process. For example, an orchestrator could deploy agents with strong recovery capabilities (high ERF) during initial evidence gathering, then switch to calibrated synthesizers (low CE) once evidence quality stabilizes, leveraging the complementary strengths identified through our framework. This metric-driven orchestration represents a practical application of our evaluation framework for building more effective multi-agent systems.

## M    EXTENDED VALIDATION: GENERALIZABILITY ACROSS DIVERSE TOOLS
##        AND MODELS

To assess the generalizability of our framework, we conducted the following validation using traces from state-of-the-art multi-tool agents on challenging tasks.

**1. Data and Trace Sampling.** We obtained traces from a diverse set of state-of-the-art agents: `Deepresearch-30B`[3], `ASearcher-32B`[4], `GPT-4o`, `GPT-5`, and `Claude Sonnet 4.5`. Each agent, equipped with web browsing, Python interpreter, and website visitation tools, was evaluated on four benchmarks: *WebWalker* (Wu et al., 2025), *GAIA* (Mialon et al., 2023), *BrowseC-*

---

[3]https://huggingface.co/Alibaba-NLP/Tongyi-DeepResearch-30B-A3B
[4]https://huggingface.co/inclusionAI/ASearcher-Web-QwQ

Table 9: Human-LLM Agreement Across Diverse Tool Types. Scores for Type and Groundedness pertain to *tool input* actions (such as search queries), while Clarity and Sufficiency scores assess the *tool output* (i.e., the produced evidence).

| Tool | $\kappa$ (Type) | $\kappa$ (Groundedness) | $\kappa$ (Clarity) | $\kappa$ (Sufficiency) |
|---|---|---|---|---|
| Web Browse / Visit | 0.80 | 0.71 | 0.68 | 0.70 |
| Code Execution | 0.77 | 0.76 | 0.73 | 0.74 |

*omp* (Wei et al., 2025), and *XBench-DeepSearch*[5]. For every combination of agent and benchmark, we randomly sampled 10 trajectories for analysis.

**2. Judge Annotation Protocol and Adaptation.** We applied our LLM-as-judge to the 200 traces with gpt-4.1-mini. The core annotation schema and prompts for reasoning types and quality attributes remained unchanged. The only adaptation was in the evidence-parsing logic: the judge was instructed to treat the direct output of any tool as the 'retrieved evidence' for the subsequent step. For example, the output of a PythonInterpreter call becomes the evidentiary basis for evaluating the groundedness of the agent's next reasoning step.

**3. Human Annotation and Results.** From the LLM-as-judge annotations, we drew a stratified random sample of 65 traces, stratified by agent (model) × tools combination to ensure proportional representation across different tool usage patterns and models. The sample size ensures a 95% confidence level with a ±10% margin of error, remaining cost-effective to annotate and evaluate our proposed schema. The sampled traces exhibited a diverse tool distribution, with search and visit operations accounting for approximately 99% of tool usage (69% and 35.2% respectively, fig. 15). Expert human annotators *independently* annotated the sampled traces, with human-LLM agreement Cohen's $\kappa > 0.65$ (Table 9).

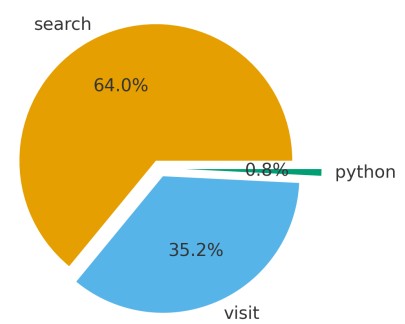

Figure 15: **Tool Distribution.** Tool usage distribution across different agents and benchmarks. Search and visit operations accounting for approximately 99% of tool usage.

**4. Stress-Testing the Judge on Ambiguous Reasoning.** We also assessed the judge's ability on ambiguous reasoning, we select an additional 20 steps where the final answer is correct but reasoning is ungrounded. For example, the agent injected "Indian" into the query despite *no geographical constraint* in the question. This represents an ungrounded inference from internal parametric knowledge, which is a critical epistemic flaw that cannot be verified.

> **Example: Ungrounded Inference from Internal Knowledge**
>
> **Question:** "An animated TV series for children first premiered in America between 2001 and 2010. The creator of this show is a physicist who obtained their PhD from a university established between 1940 and 1960... Name the reviewer."
> **Agent's Search Query:** "Indian physicist created animated series America 2000s"
> **Key Issue:** The agent injected "Indian" into the query despite *no geographical constraint* in the question. This represents an ungrounded inference from internal parametric knowledge, which is a critical epistemic flaw that cannot be verified.
> **Label:** LLM judge and human annotator both label this as "Not Grounded".

Re-evaluation showed that human-LLM agreement remained acceptable ($\kappa \approx 0.70$ for groundedness, $\kappa \approx 0.66$ for clarity, $\kappa \approx 0.65$ for sufficiency), confirming our framework's robustness in exposing even nuanced reasoning failures.

---

[5]https://huggingface.co/datasets/xbench/DeepSearch

# N    USE OF LARGE LANGUAGE MODELS

Large Language Models (LLMs) played a significant role in this research, warranting disclosure of their contributions. LLMs were extensively used as annotation helpers to develop our epistemic competency schema, as analytical assistants for reasoning trace analysis, and as automated judges (detailed in Section D). They contributed significantly to the writing process by generating initial drafts of technical sections and assisting with revision, and influenced research ideation by suggesting evaluation metrics and identifying methodology gaps. All LLM-generated content underwent rigorous human review and validation, with human authors verifying analyses, validating schemas through expert review, and thoroughly editing all contributions. While LLMs served as powerful assistive tools, all final decisions regarding research direction, experimental design, result interpretation, and manuscript content were made by human authors.

