# OpenReview forum: "Do LLM Agents Know How to Ground,  Recover, and Assess? Evaluating Epistemic Competence in Information-Seeking Agents"
_ICLR.cc/2026/Conference — ICLR 2026 Poster_

### Official Review · Reviewer_ESgZ · 2025-10-27

**Soundness:** 3
**Presentation:** 3
**Contribution:** 3
**Rating:** 6
**Confidence:** 3

**Summary:**

This paper proposes SeekBench, a benchmark to evaluate LLM search agents on epistemic competence of three dimensions/metrics: reasoning groundedness, search recovery, and answer calibration.

**Strengths:**

1. The proposed benchmark is well-designed, and the methodology is well-elaborated.
2. The evaluation is extensive on seven QA datasets, using multiple LLM search agents.
3. Further analysis provides insights into the differences between LLM agents regarding the proposed metrics/capabilities.

**Weaknesses:**

- Some concerns regarding methodology, experiments, and analysis. Please refer to "Questions" below.

**Questions:**

1. What are the raw data source(s) and domain(s) for constructing SeekBench?
2. What does each annotated feature in Table 1 mean?
3. To clarify, the human/LLM annotators will label the correctness of each trace i, as well as the clarity and quality of the retrieved evidence at each turn t in each trace, right? What exactly is "quality" defined here?
4. Section 3.3.3 Definition 3.5 (Calibration Error (CE)): Why should the ideal policy answer if and only if the evidence is sufficient? Whether the explicit evidence is sufficient or not also depends on the internal knowledge of the policy, so it would also be acceptable if the policy answers correctly without much retrieval, especially for easy questions.
5. Section 4 "We evaluate the agents on a diverse set of seven question-answering benchmarks": Does this mean the original QA datasets are transformed as SeekBench-style benchmarks using the methodology (annotation process) described in Section 3? If so, as many QA datasets do not provide reasoning traces, where do the original traces and steps come from? Why not evaluate the agents on the proposed SeekBench ("SeekBench comprises 190 expert-annotated traces with over 1,800 response steps")?
6. Section 4 analyzes the differences between LLM agents regarding the proposed metrics/capabilities, but what causes such differences? Is it because of the training schema or other aspects of different LLM agents? What do the comparisons reveal? What are the insights into improving the current LLM agents based on the findings of SeekBench?

---

> ### Author Response · Authors · 2025-11-21
> **Rebuttal Q1-Q4**
>
> We sincerely thank the reviewers for their insightful and constructive comments and questions. We have carefully considered each question and revised the manuscript accordingly.
>
> `Q1. What are the raw data source(s) and domain(s) for constructing SeekBench?`
> ## R.Q1 - Data Sources and Domains (Revised Appendix C.1)
>   - Raw data sources: Seven question-answering benchmarks: NQ, TriviaQA, PopQA (single-hop), HotpotQA, 2Wiki, MusiQue, and Bamboogle (multi-hop).
>   - Domain: Open-domain question answering.
>
> `Q2. What does each annotated feature in Table 1 mean?`
> ## R.Q2.
> Thanks to the reviewer's comments, we now provide a **more comprehensive Figure 9** to illustrate the annotated feature.
> Our annotation schema consists of **annotated features** organized into two dimensions:
>
> ## A. Functional Types (Cognitive Purpose)
>
> ### A.1 Reasoning Types:
> - **InformationSynthesis**: Synthesizes information from search results to form a conclusion (e.g., "Based on CNN and BBC reports, Biden won with 306 electoral votes").
> - **PlanFormation**: Forms a plan of action for future searches (e.g., "I should search for 2020 US election results").
> - **StateAssessment**: Assesses the current knowledge state, usually identifying a knowledge gap (e.g., "I don't know who won the 2020 election yet").
>
> ### A.2 Search Types:
> - **InitialQuery**: The first query in a reasoning chain.
> - **RefinedQuery**: Refines a previous query based on new information (e.g., adding "electoral college" for precision).
> - **FollowUpQuery**: A follow-up question that is not a direct refinement (e.g., asking about voter turnout when the original question was about the winner).
> - **RepeatQuery**: A query that is the same as the previous query.
>
> ## B. Quality Attributes (Epistemic Soundness)
>
> ### B.1 For Reasoning Steps:
> - **grounding**: Binary label indicating whether all factual premises in the reasoning step are supported by explicit evidence spans in the retrieved information.
>   - **Grounded**: ALL atomic premises have direct supporting evidence.
>   - **Not Grounded**: ANY premise lacks evidence support, or the step contains only meta/plan text without factual claims.
>
> ### B.2 For Search Results:
> - **quality** (sufficiency): Whether the search result contains enough information to answer the query.
>   - **Sufficient**: The answer appears to be present in the results.
>   - **Insufficient**: The answer is likely not present.
> - **clarity**: Whether the information is unambiguous and interpretable.
>   - **Clear**: Information is straightforward and addresses one subject.
>   - **Unclear**: Results mention multiple entities that could match the query, or the information is vague/ambiguous.
>
> ### B.3 For Final Answers:
> - **correct**: Binary label indicating whether the agent's final answer is correct or incorrect according to ground truth.
>
> These features enable systematic evaluation of three epistemic competencies: **Groundedness** (via grounding+evidence state), **Recovery** (via search types+evidence state), and **Calibration** (via answer correctness+evidence state).
>
> `Q3. To clarify, the human/LLM annotators will label the correctness of each trace i, as well as the clarity and quality of the retrieved evidence at each turn t in each trace, right? What exactly is "quality" defined here?`
>
> ## R.Q3.
> - Yes, human/LLM annotators will label the correctness of each trace i, as well as the clarity and quality of the retrieved evidence at each turn t in each trace.
> - `What exactly is "quality" defined here?` We added a **clarifying sentence** in Definition 3.1 explicitly stating that **"quality" means "sufficiency"** to avoid confusion with general quality assessments.
>
> `Q4. Section 3.3.3 Definition 3.5 (Calibration Error (CE)): Why should the ideal policy answer if and only if the evidence is sufficient? Whether the explicit evidence is sufficient or not also depends on the internal knowledge of the policy, so it would also be acceptable if the policy answers correctly without much retrieval, especially for easy questions.`
>
> ## R.Q4
>   - Search agents are defined as agents that **interact with external knowledge sources** to answer given questions. We revised **Introduction L37-39** to explicitly state the definition.
>   - **Data Sanitization**: In our evaluation, we removed questions answerable by internal knowledge alone (Section 4, L375 + Appendix B).
>
>   `Definition 3.5 (Calibration Error (CE)): Why should the ideal policy answer if and only if the evidence is sufficient?`
>
> - Ideal policy definition (Definition 3.5): Answers if and only if evidence is sufficient ($E_{i,t} = 2$). Our evaluation focuses on **retrieval-dependent** reasoning scenarios, i.e., internal knowledge is insufficient, answer quality depends entirely on the retrieved evidence.
> - Answering with insufficient evidence → overconfidence (high error rate)
> - Not answering with sufficient evidence → overcautiousness (wasted effort)
> - Therefore: ideal policy = answer if and only if evidence is sufficient

---

> ### Author Response · Authors · 2025-11-21
> **Rebuttal Q5 & Q6**
>
> `Q5-1 Does this mean the original QA datasets are transformed as SeekBench-style benchmarks using the methodology (annotation process) described in Section 3? If so, as many QA datasets do not provide reasoning traces, where do the original traces and steps come from?`
> ## R.Q5-1
>   - QA benchmarks only provide questions (and answers), not reasoning traces. The traces are generated when agents run on these benchmarks.
> | Step || Description|
> |---|---|----|
> |1| Agents receive questions from QA benchmarks| Benchmarks provide *only* questions (and gold answers) — **no reasoning traces** are included.|
> |2|Agents generate multi-turn traces| While solving the questions, agents produce their own reasoning: search steps, evidence retrieval, intermediate thoughts, and answer formulation. |
> |3| We apply our annotation schema| Our framework labels the agent-generated traces and evaluates them using quantitative metrics.|
> - We do not transform QA datasets into SeekBench-style benchmarks. Instead, we annotate agent-generated traces using our epistemic competence framework.
> - We have **revised Appendix A** to clarify this data selection (process).
>
>  `Q5-2 Why not evaluate the agents on the proposed SeekBench ("SeekBench comprises 190 expert-annotated traces with over 1,800 response steps")?`
> ## R.Q5-2
> - The 190 traces serve as a "validation set" to establish framework reliability as mentioned in **Introduction L87-90**. We establish annotation schema reliability (κ > 0.8) and validate LLM-as-judge feasibility (κ > 0.7)
> - The 28,493 traces serve as the "test set"(large-scale evaluation) for comprehensive evaluation.
> - This "validation set" + LLM for scalability is widely applied [1][2].
>
> [1] Zhang et al., Which Agent Causes Task Failures and When? ICML 2025
>
> [2] Cemri et al., Why Do Multi-Agent LLM Systems Fail?, NeurIPS 2025
>
> `Q6. Section 4 analyzes the differences between LLM agents regarding the proposed metrics/capabilities...`
>
> ## R.Q6
>   We thank the reviewer for their feedback. We have **revised Section 4.5 L510** and **Appendix I & J** to discuss our findings on training objective design and broaden our analysis of agent differences and improvement insights. Specifically,
>
> `Q6-1 what causes such differences? Is it because of the training schema or other aspects of different LLM agents?`
>
>  ## R.Q6-1
> - Training objective design is the primary factor driving epistemic competence differences. For example,
> - Search-R1: Optimizes only for final answer correctness → achieves high accuracy but sacrifices reasoning groundedness
> - ASearcher: Emphasizes **data synthesis** for high-quality data with failure modes and recovery paths to teach the agent how to re-plan → excels in evidence acquisition and recovery (aligned with our Recovery analysis Section 4.3)
>
> `Q6-2 What do the comparisons reveal?`
> ## R.Q6-2
> 1. Section 4.2: Agent's reasoning groundedness is obscured even when answer is correct. For example,
> | Component| Content| Notes|
> |---|---|---|
> | **Question** | *What is the largest planet in our solar system?*|  |
> | **Retrieved Evidence** | *"Jupiter is a gas giant. Planets vary in size."*| Does **not** state that Jupiter is the largest |
> | **Agent’s Reasoning** | *"The evidence mentions Jupiter as a gas giant. Gas giants are typically large, so Jupiter is probably the largest planet."* | Reasoning **not grounded** in evidence |
> | **Final Answer**     | **Jupiter** ✓| Correct answer, but unsupported by evidence    |
> (We also added a motivating example **Figure 1** to illustrate)
>
> 2. Section 4.3: RL training improves calibration → shows competency-specific effects of training objectives.
>    - As we have discussed in Appendix G: Base model has lowest F1 (33.5%) independently, but other models (better calibration) can achieve +3.50 F1 using Base's evidence.
>    - Base collects high-quality evidence but struggles with synthesis, missed by accuracy-only evaluation.
> 3. Section 4.4: Agent specialization (Search-R1's synthesis) → demonstrates how different training approaches produce different epistemic competencies
>
> `Q6-3 What are the insights into improving the current LLM agents based on the findings of SeekBench?`
> ## R.Q6-3
> 1. Outcome-only optimization is insufficient: Optimizing only for final answer accuracy (as in Search-R1) fails to develop evidence-grounded reasoning (Figure 3)
> 2. Agent specialization can be combined: Different training methods produce complementary strengths that can be leveraged through modular architectures (Table 7)
> 3. We augmented ASearcher-7B with **inference-time feedback** signals derived from our three epistemic competencies (**added Appendix J**).
>     - It achieves an 8.4% increase in final F1 score, with improvements across all three competencies: groundedness (RQI) +13.3%, recovery (ERF) +6.5%, and calibration error (CE) -5.8%.
>     - This demonstrates that our framework provides **actionable signals** that directly improve both process-level reasoning and final answer quality.

---

> ### Comment · Reviewer_ESgZ · 2025-11-28
> **Rebuttal Acknowledgement by Reviewer ESgZ**
>
> Thanks for the clarification and additional details regarding the raised questions, especially the insightful discussion about Q6, which also adds to the significance of the proposed benchmark and points out future directions on how to improve models.
>
> I would **rate this paper as 7**, although this option is not available.

---

### Official Review · Reviewer_9ECN · 2025-10-28

**Soundness:** 3
**Presentation:** 2
**Contribution:** 3
**Rating:** 4
**Confidence:** 3

**Summary:**

This paper proposes SeekBench, an evaluation framework for analyzing the intermediate reasoning and search process of search-based LLM agents. The authors construct a framework centered on 3 competencies (groundedness, recovery, and calibration), proposing various cases of each competencies and associated metrics.

**Strengths:**

- Evaluation beyond outcome is an important topic: Validating that the processes search agents take in finding the final answer is necessary for reliability, transparency, and can be a valuable feedback signal for improving future agents.
- For search-based agents, the creation of the competencies, annotated features, and metrics seems reasonable, involving three rounds of review with humans and LLM-as-judges. This results in an evaluation schema that is both principled and scalable, with automatic evaluators exhibiting high levels of aggregate-agreement with humans.
- The analysis identified clear shortcomings of RL-trained search agents: outcome-based RL training leads to better calibrated agents (i.e., agents tend to produce answers when the evidence state is both clear and sufficient to answer the question). However, the intermediate reasoning steps produced by agents are not necessarily grounded, reflecting the outcome-driven nature of RL training.

**Weaknesses:**

- In my opinion, many parts of the abstract and introductory sections are insufficiently clear, and largely have to do with the repeated statement that SeekBench is a process-level benchmark consisting of 190 traces with 1800+ annotated steps (i.e., in abstract, L88, L97). This framing lead me to believe that SeekBench was a benchmark for evaluating process-level evaluators, like those present in math domains, e.g., ProcessBench. **In reality, an annotation schema is derived from said traces, and used to validate automatic evaluation approaches, like LLM-as-judges**. This should be made clearer to the reader, especially as the evaluation process in Section 4 does not really directly use these annotations.
- The paper seems to only analyze search agents that can use a single web-search tool. This may not be representative of state-of-the-art search agents, e.g., [1,2], which are capable of multiple tools, such as using a diverse array of tools like code interpreters or more complex forms of search like web browsing. It’s unclear how SeekBench analysis holds when agents are given toolkits more representative of real-world use-cases. I don’t think states with other tools can easily be folded into existing competencies, nor do I think the existing framework can be applied directly to other tools.
- Evaluated baselines:
  - It seems that only agents of 7B model size are evaluated, even though many baselines have larger, more capable model variants. For example, ASearcher has 14B and 32B variants, trained from Qwen2.5-14B and QwQ-32B, respectively.
  - Furthermore, newer, more capable tool-calling baselines, such as WebSailor [2] (and v2 variant) are missing. These missing baselines may be a result of the above weakness, as they are typically trained to use multiple tools.
  - Lastly, API models with search tool-calling ability (e.g., OpenAI models) are not analyzed.
  - This weakness is critical in my opinion. Because this paper mainly performs analysis on the behavior of search-based agents, it should aim to cover as many search agents are reasonably possible. But a critical dimension, agent model size, is missing! How many of the unearthed agent shortcomings can be mitigated or solved simply by scaling up the agent size?


[1] https://arxiv.org/abs/2509.06283

[2] https://arxiv.org/abs/2507.02592

**Questions:**

- LLM-as-judge agreement with human annotations is presented at an aggregate level, without per-metric breakdowns. What is the per-metric breakdown? Are judges more reliable for certain metrics? If yes, can we still reliably trust judges for **all** metrics?
- What is approximate cost of annotating one search agent in terms of LLM-as-judge API calls? It seems that each intermediate step in an agent requires multiple API calls.
- The authors have a unique opportunity to more thoroughly benchmark LLM-as-a-judge models on the human annotations to determine the best model based on cost-performance tradeoffs. Why were GPT-4.1, 4.1-mini, and GPT-5 chosen? Why not evaluate GPT-5-mini,nano, or models like Claude Sonnet, Gemini 2.5 Pro, or more capable local models that are “free” (in terms of API cost)? Further, what reasoning effort was GPT-5 evaluated with? To inform practitioners, how does this performance vary with reasoning effort setting?

---

> ### Author Response · Authors · 2025-11-21
> **Rebuttal W1 & W2**
>
> We sincerely thank reviewer's helpful comments. We have carefully considered each suggestion and revised the manuscript accordingly.
>
> ## W1
> `...insufficiently clear, and largely have to do with the repeated statement that SeekBench is a process-level benchmark...lead me to believe that SeekBench was a benchmark for evaluating process-level evaluators...In reality, an annotation schema is derived from said traces, and used to validate automatic evaluation approaches, like LLM-as-judges....`
>
> ## R1
>   We thank the reviewer for pointing out this confusion. We have revised the Abstract and Introduction to clarify that:
>   1. SeekBench contributes
>   (i) an annotation schema for search-agent traces, and
>   (ii) a curated set of 190 human-annotated traces (1,800+ steps) used to **validate the schema** and to calibrate LLM-as-judge–based automatic evaluation.
>   2. Section 4 benchmarks systems using LLM-as-judge guided by this schema; the human annotations are used only to validate evaluator reliability and provide analyses, *not* to score models.
>
> ## W2
>
> `The paper seems to only analyze search agents that can use a single web-search tool. This may not be representative of state-of-the-art search agents, e.g., [1,2], which are capable of multiple tools, such as using a diverse array of tools like code interpreters or more complex forms of search like web browsing. It’s unclear how SeekBench analysis holds when agents are given toolkits more representative of real-world use-cases. I don’t think states with other tools can easily be folded into existing competencies, nor do I think the existing framework can be applied directly to other tools.`
>
> ## R2.
> We respectfully note that the reviewer's concern is based on claims our paper **never makes**. Our paper never states that:
> - "states with other tools can easily be folded into existing competencies"
> - "the existing framework can be applied directly to other tools"
> - We appreciate the opportunity to clarify our actual research scope and contributions, but the above concerns are not valid criticism of our work.
>
> ---
> Our framework intentionally focuses on epistemic evaluation by design (**Revised L46-50**)
> 1. Textual search queries present a **unique epistemic evaluation** challenge.
>   - Unlike code (executable) or mathematics (logically verifiable), search query quality cannot be objectively determined—it depends on the interplay between the query, retrieved evidence, and reasoning process.
>   - This epistemic nature is precisely what makes it an **unsolved evaluation** problem worthy of dedicated study.
> 2. Contribution: Our work establishes the first systematic framework for evaluating epistemic competence in search agents. By focusing on textual search, we can develop a rigorous, validated framework with: High inter-annotator agreement (κ > 0.8); Strong LLM-as-judge alignment (κ > 0.7); Formal mathematical definitions of epistemic competencies; and Large-scale validation.
> 4. We appreciate reviewers mentioning two baselines WebSailor, and SFR-DeepResearch. We have **included WebSailor and ASearcher-Web**, but SFR-DeepResearch is **not** open-sourced yet. Please check our response to "Evaluated baselines" (W3) for detailed results.
>
> ---
> The reviewer's concern about "representativeness" conflates tool diversity with research validity.
> - Our research question is: “How do we evaluate the epistemic competence of LLM agents in information-seeking scenarios?”
> - Evaluating agents with textual search tools is representative *for our research question*.
>   - A study on "mathematical reasoning ability" that focuses on mathematical problem-solving is representative for that research question, even if it **doesn't include** calculators, graphing tools, or other mathematical aids. The research question defines the appropriate scope. For example: Wang et al., *Beyond Accuracy: Dissecting Mathematical Reasoning for LLMs Under Reinforcement Learning, NeurIPS 2025*
> - Finally, we have clearly delineated our research boundaries in revised version [L46-82] where we Explicitly state textual search only for evaluating epistemic competence (evidence-driven).

---

> ### Author Response · Authors · 2025-11-21
> **Rebuttal W3 (baselines)**
>
> ## W3. Evaluated baselines:
>   `It seems that only agents of 7B model size are evaluated, even though many baselines have larger, more capable model variants. .....`
>  `Furthermore, newer, more capable tool-calling baselines, such as WebSailor [2] (and v2 variant) are missing....`
> `Lastly, API models with search tool-calling ability (e.g., OpenAI models) are not analyzed.`
>
> `This weakness is critical in my opinion. Because this paper mainly performs analysis on the behavior of search-based agents, it should aim to cover as many search agents are reasonably possible. But a critical dimension, agent model size, is missing! How many of the unearthed agent shortcomings can be mitigated or solved simply by scaling up the agent size?`
>
> ## R3.1 Clarfication
>    - Our primary evaluation focuses on **state-of-the-art search agents** (Search-R1, ReSearch, ASearcher, DeepResearcher). These represent the current research community's main focus in **RL-trained** agents.
>   - Our research goal: Establish the first systematic evaluation framework for **epistemic competence**—a methodological contribution that provides assessment tools, **not** a study of how **model size** affects performance.
>   - Our framework evaluates process-level behaviors (groundedness, recovery, calibration) that are independent of model parameters.
>    - Our contribution is the **evaluation framework** itself. It can assess epistemic competence for any model size, just as a thermometer measures temperature regardless of the object's size.
>   - **Existing work on agentic behaviors often doesn't consider mitigating some behavior through model size**!
>     - Zhang et al., Which Agent Causes Task Failures and When? ICML 2025 → Only considers GPT-4o-based agents.
>     - Cemri et al., Why Do Multi-Agent LLM Systems Fail?, NeurIPS 2025 → Only consider agents built on top of GPT-4o and Claude 3.
>     - The focus of our paper aligns with existing literature.
>   - In response to this concern, we evaluated ASearcher, WebSailor (7B & 32B variants) and GPT-5-mini on GAIA for web searching. However, this analysis serves to demonstrate framework applicability across sizes, **not to study scaling effects**.
>   - Our contribution remains the **evaluation methodology itself**, which reveals epistemic competence patterns regardless of model size.
>   - The value of our framework lies in **making epistemic behaviors measurable and comparable**, not in comparing model capacities. Whether limitations persist across sizes is a separate research question about scaling effects, **not a limitation of our evaluation methodology**.
>
> ## R3.2 ASearcher & WebSailor & GPT Included for Web Searching
>
> We have evaluated WebSailor and ASearcher (web browsing + visit capabilities) on GAIA and included results in the revision (**Appendix K**).
>
> *Note that only ASearcher and WebSailor have *web-search* variants, and WebSailor does not open source 14B.*
>
> The results for 7B scale and 32B scale are as follows:
>
> ---
> - Answer-Level Performance (Pass@2)
>
> | Model | Scale | Pass@2 (%) |
> |---|---|---|
> | **ASearcher** | 7B /32B | 16.5 /29.2 |
> | **WebSailor** | 7B /32B | **18.2** /**32.4** |
> | **GPT-5-mini** | - | 15.7 |
>
> - We also add inference-time feedback to GPT-5-mini with our LLM-as-judge framework (Appendix J). It achieved Pass@2 22.5% (+43% rel.). This demonstrates that our framework provides actionable signals to improve final answer.
>
> ### Comparison
> | Model | RQI (Groundedness) ↑ | ERF (Recovery by Turn 8) ↑ | CE (Calibration Error) ↓ |
> |---|---|---|---|
> |**GPT-5-mini**|**0.45**|42%|0.34|
> | **ASearcher-7B** | **0.22** | **42%** | 0.38 |
> | **WebSailor-7B** | 0.15 | 35% | **0.33** |
> | **ASearcher-32B** | **0.28** | **58%** | 0.35 |
> | **WebSailor-32B** | 0.19 | 45% | **0.31** |
>
> - **ASearcher** maintains superior **groundedness** (RQI = 0.28) and **recovery** (ERF = 58%)
> - **WebSailor** continues to show better **calibration** (CE = 0.31)
>
> ### Cross-Scale Performance Trends
>
> | Metric | ASearcher (7B→32B) | WebSailor (7B→32B) |
> |---|---|---|
> | **Pass@2** | +77% rel. |+78% rel. |
> | **RQI** |+27% | +27%|
> | **ERF** | +38%  | +29% |
> | **CE** | -8% | -6% |
>
> - Both models show improvements across all metrics when scaling from 7B to 32B
> - **ASearcher** shows larger absolute gains in recovery
> - **WebSailor** maintains its calibration advantage at both scales
> - Calibration error decreases for both models with scale, indicating **better** evidence-aligned decision-making
>
>
> ### Tool Selection Follows Epistemic Scope
>   - Web browsing/visit (WebSailor): Included, as these require epistemic evaluation—quality depends on reasoning and evidence, not objective correctness.
>   - Code interpreter (WebSailor v2 variant): Excluded, as these are objectively verifiable (clear right/wrong); not within our epistemic study. **This is a methodological choice, not a limitation**.

---

> > ### Author Response · Authors · 2025-11-21
> > **Rebuttal Questions**
> >
> > `Q1 LLM-as-judge agreement with human annotations is presented at an aggregate level, without per-metric breakdowns. What is the per-metric breakdown? Are judges more reliable for certain metrics? If yes, can we still reliably trust judges for all metrics?`
> > ## R.Q1
> > We already reported per-metric breakdowns in **Appendix A, Fgiure 6** of the original version. In the revised version, we also report that for new LLM models.
> > - Across all LLM judges, per-metric IAA scores are consistently > 0.6. Specifically, GPT-4.1-mini, the most cost-effective judge, achieves per-metric κ ≥ 0.67.
> >
> > `Q2. What is approximate cost of annotating one search agent in terms of LLM-as-judge API calls? It seems that each intermediate step in an agent requires multiple API calls. `
> > ## R.Q2
> > We thank the reviewer's question on different LLM models as judges. Although our main focus is **not on benchmarking every available model**, we have revised Appendix A (Pareto frontier analysis, Figure 7) to show GPT-4.1-mini is Pareto optimal and cost-effective.
> > | Model| IAA (κ ± std) | Token Cost / trace ($) | Time Cost / trace (s) |
> > | ----------------- | ------------- | ---------------------- | --------------------- |
> > | **GPT-4.1-mini**  | 0.731 ± 0.057 | **$0.0087**| **2.48 s**  |
> > | GPT-5-nano        | 0.683 ± 0.038 | $0.0098  | 4.56 s |
> > | Gemini-2.5-flash  | 0.643 ± 0.102 | $0.0238                | 4.77 s                |
> > | GPT-4.1           | 0.693 ± 0.043 | $0.0441                | 1.47 s                |
> > | GPT-5-mini        | 0.791 ± 0.028 | $0.0485                | 5.26 s                |
> > | GPT-5             | 0.754 ± 0.032 | $0.1016                | 28.47 s               |
> >
> > `Q3. The authors have a unique opportunity to more thoroughly benchmark LLM-as-a-judge models on the human annotations to determine the best model based on cost-performance tradeoffs.  Why were GPT-4.1, 4.1-mini, and GPT-5 chosen? Why not evaluate GPT-5-mini,nano, or models like Claude Sonnet, Gemini 2.5 Pro, or more capable local models that are “free” (in terms of API cost)?`
> >
> > ## R.Q3
> >  We thank the reviewer for their valuable question and appreciate the opportunity to make our selection more rigorous.
> >   - Although **most works directly** use proprietary state-of-the-art models, e.g., Cemri et al.[a], we have tested GPT-5-mini, GPT-5-nano, and Gemini-2.5-flash and identified the models (GPT-4.1-mini and GPT-5) that lie on the **Pareto frontier**—those offer the best trade-off between annotation quality (IAA) and cost (token & time) (see Appendix A).
> >     - GPT-5-mini is 5.6× more expensive than GPT-4.1-mini for only a marginal IAA increase (+0.06).
> >     - Gemini-2.5-flash had lower IAA (0.643) than GPT-4.1-mini (0.731), at nearly 3× the cost.
> >
> >   [a] Why Do Multi-Agent LLM Systems Fail?, NeurIPS 2025
> >
> >   `Why not evaluate models like Claude Sonnet, Gemini 2.5 Pro`
> >   - Claude Sonnet and Gemini 2.5 Pro were not tested because both of them output **tokens cost higher** than GPT-5 ( \\$10/M ) [claude-sonnet-4 costs \\$15/M https://docs.claude.com/en/docs/about-claude/pricing, gemini-2.5-pro costs at least \$10/M depends on sequence length https://ai.google.dev/gemini-api/docs/pricing]. These models are **neither affordable nor scalable for large-scale annotation tasks**.
> >
> >  `Why not evaluate more capable local models`
> >   - Top-tier open models (<= 32B) like Qwen3-30B-A3B [see leaderboard https://lmarena.ai/leaderboard/text] require **~4×A100 GPUs** for inference to match throughput and concurrency—unrealistic for scalable deployment!!
> >     - One A100 costs ~\$16,000; running costs and setup are non-trivial.
> >     - Throughput and latency are inferior to API-based models.
> >     - By contrast, GPT-4.1-mini costs ~$0.0087 per trace via API—zero hardware overhead, infinite elasticity, and reproducibility.
> >   - Recent study [1] has shown that using open-source local models for LLM-as-judge, even with fine-tuning, still performs worse. Fine-tuned open-source models not only need high quality data from GPT, but high computation resources (8*A100) [2].
> >     - [1] Huang et al. , An Empirical Study of LLM-as-a-Judge for LLM Evaluation: Fine-tuned Judge Model is not a General Substitute for GPT-4, ACL 2025
> >     - [2] Zhu et al., Judgelm: Fine-tuned large language models are scalable judges, ICLR 2025.
> >
> >  `Further, what reasoning effort was GPT-5 evaluated with? To inform practitioners, how does this performance vary with reasoning effort setting?`
> >
> > Our paper uses gpt-5-nothinking. As shown in Appendix A, even GPT-5 (nothinking) is **not Pareto-optimal**, with over 11× more expensive than GPT-4.1-mini. GPT-5-thinking is **internally excluded** because the thinking process further amplifies inefficiency. To respond, we evaluate gpt-5-thinking:
> >
> > | **Model** | **Time per Trace (s)** | **Cost per Trace (\$)** | **IAA**  |
> > |--------------------|-------------------------|---------------------------|----------|
> > | GPT-5-thinking| 30.20 | 0.12| 0.757    |
> > | GPT-5 |   28.47| 0.10 | 0.754    |

---

> > > ### Comment · Reviewer_9ECN · 2025-11-25
> > > **Response to authors, pt 1**
> > >
> > > Thanks to the authors for their extensive comments and new experiments. I think the revisions have clarified the contributions. I have some follow-ups.
> > >
> > > > We respectfully note that the reviewer's concern is based on claims our paper never makes. Our paper never states [...]
> > >
> > > I believe the authors have misunderstood my concern. I never stated that you should evaluate coding agents or math reasoning models, nor did I say that your work claims that other tools can be accommodated in the analysis framework.
> > >
> > > Rather, my concern is that *if* one uses your framework to assess an agent that uses tools beyond just search, can one deem the evaluation results reliable given your human judgment studies? As my review suggests, changes to metrics and judge prompts are required, but these changes mean that the correlation study between judges and human labels, which were conducted based on trajectories from search-only agents, no longer fully apply.
> > >
> > > One may argue that this framework should only be applied to agents only capable of search. Then, it seems that the utility of this framework is somewhat limited: Search-only agents constitute a decreasing share of information-seeking agents, as SOTA agents are increasingly equipped with richer toolkits; see the recent releases of Tongyi DeepResearch, MiroThinker, and DR-Tulu, for example.
> > >
> > > > ...Textual search queries present a unique epistemic evaluation challenge...Unlike code (executable) or mathematics (logically verifiable), search query quality cannot be objectively determined—it depends on the interplay between the query, retrieved evidence, and reasoning process.
> > > This epistemic nature is precisely what makes it an unsolved evaluation problem worthy of dedicated study.
> > >
> > > > The reviewer's concern about "representativeness" conflates tool diversity with research validity.
> > >
> > > Other tools, like web browsing, clicking, or file opening/processing, are similar to web search: they depend on the query, the existing context (retrieved evidence), and the reasoning process. Is it reasonable to claim to be "the first systematic evaluation framework for epistemic competence" when the framework is not validated for even a reasonable set of tools that agents use for information gathering?
> > >
> > > Further, I don't think that code or math evaluation necessarily are more easy to evaluate, as the authors seem to suggest. Simply because code executes does not mean the intention or inputs of the code snippet are correct. In the context of information-seeking agents, an agent still needs to (1) write the code, (2) ensure code inputs are correct, as derived from retrieved evidence and corresponding reasoning, and (3) reason about the outputs of the code snippet.
> > >
> > > > Our contribution remains the evaluation methodology itself, which reveals epistemic competence patterns regardless of model size...The value of our framework lies in making epistemic behaviors measurable and comparable, not in comparing model capacities. Whether limitations persist across sizes is a separate research question about scaling effects, not a limitation of our evaluation methodology.
> > >
> > > The authors still claim "large scale analysis" as a contribution in their paper, both in abstract and in introduction. Even the title calls this work a "benchmark". It is then natural to evaluate the work based on how relevant and thorough the analysis and benchmarking efforts are. The abstract and intro assert generalized takeaways like "Our evaluation [...] reveals that RL agents excel at evidence gathering but struggle with reasoning". Can one reasonably assert this without investigating the effects of model scale?
> > >
> > > Related: Given that the 190 human annotated traces are generated from weaker (7B) agents, can the authors guarantee reliability of their system in evaluating larger models? That is, is there evidence that traces from 7B models are representative of errors made by agents of larger sizes? Recent works [1,2] analyzing LLM behavior in reasoning domains seem to indicate more capable LLMs make harder-to-catch mistakes. Do we have evidence that information-seeking agents are different?
> > >
> > > [1] https://arxiv.org/abs/2501.15581
> > > [2] https://arxiv.org/abs/2509.17995

---

> > > > ### Comment · Reviewer_9ECN · 2025-11-25
> > > > **Response to authors, pt 2**
> > > >
> > > > > Top-tier open models (<= 32B) like Qwen3-30B-A3B require ~4×A100 GPUs for inference to match throughput and concurrency—unrealistic for scalable deployment!! ... One A100 costs ~16,000; running costs and setup are non-trivial... Throughput and latency are inferior to API-based models. By contrast, GPT-4.1-mini costs ~0.0087 per trace via API—zero hardware overhead, infinite elasticity, and reproducibility.
> > > >
> > > > I'm not saying hardware is free, but there are practical reasons why a user may prefer locally hosted evaluators. For example, enterprise users may not want to send proprietary information to 3rd party APIs.
> > > >
> > > > Lastly, I do disagree with the claim that relying on API-based inference is more reproducible. API-based models are black-boxes: models backends may be quantized or modified without full transparency.
> > > >
> > > > > Recent study [1] has shown that using open-source local models for LLM-as-judge, even with fine-tuning, still performs worse. Fine-tuned open-source models not only need high quality data from GPT...
> > > >
> > > > I am quite familiar with the LLM-as-judge finetuning space. The baselines used in [1] are quite weak by judge standards (largely late 2023/early 2024 models), and are not representative of state-of-the-art finetuned judges. Finetuning methods have progressed beyond simple distillations from GPT-4.

---

> > > > ### Author Response · Authors · 2025-11-25
> > > > **Response pt1 (1/3)**
> > > >
> > > > We thank the reviewer for their thoughtful and detailed feedback! Below, we address the key concerns point-by-point, and clarify our claims, scope, and intended contributions.
> > > >
> > > > ### **1. Scope and Tool Diversity**
> > > >
> > > > > *“If one uses your framework to assess agents that use tools beyond just search, can one deem the evaluation results reliable given your human judgment studies?”*
> > > >
> > > > - We agree that modern agents increasingly use richer toolsets (e.g., browsing, scraping, file parsing, Python execution). However, we respectfully clarify that our framework evaluates **epistemic behaviors**—not the surface API or tool modality.
> > > >
> > > > - Our contribution lies in defining **process-level epistemic competencies** (groundedness, recovery, and calibration) **over the agent’s interaction with evidence**, regardless of **how that evidence is acquired**.
> > > >
> > > > - While our empirical validation focuses on **search-based agents**, the **epistemic structure** remains across diverse tools, i.e.,
> > > > `Agent proposes evidence-seeking action → obtains external evidence → reasons over it → decides next action → answers`. Our framework targets this process, independent of the surface tool API, to maintain precise empirical scope (**revised *Introduction, L42***).
> > > >
> > > > #### **Examples:**
> > > >
> > > > * In **MiroThinker**, the `scrape_and_extract_info` tool uses a lightweight LLM to extract relevant spans from webpages. Though the tool is more expressive than basic search, the agent behavior follows the same epistemic loop: `detect information gap → retrieve → reason → recover`. Our metrics apply identically.
> > > >
> > > > * In **DR-Tulu**, the `browse` tool replaces snippet-level search with page-level summaries. Again, the epistemic failures (unsupported inference, premature answer, lack of refinement) remain unchanged.
> > > >
> > > > * In **DeepResearch**, the `PythonInterpreter` tool executes code. The output (e.g., `"result": 455000`) becomes **evidence**. The agent must:
> > > >
> > > >   1. correctly derive code inputs from context,
> > > >   2. interpret the output in relation to the query,
> > > >   3. determine if further retrieval or refinement is needed.
> > > >
> > > > These steps map directly to **groundedness**, **recovery**, and **calibration**. Thus, even tools that *generate* evidence do not alter the underlying epistemic evaluation targets.
> > > >
> > > > ---
> > > >
> > > > > *“Changes to metrics and judge prompts are required, and the correlation study... no longer fully apply.”*
> > > >
> > > > - We acknowledge that real-world agents increasingly employ diverse tools, including web scraping, PDF/file parsing, code execution, shell commands, and multimodal input. It is indeed impractical to exhaustively enumerate or hard-code support for every possible tool and its output format/evidence source.
> > > > - However, the **core epistemic labels and their behavioral manifestations** remain the same.
> > > >    - Whether an agent retrieves evidence via a search query, a Python computation, a file parse, or a structured scrape, the resulting information becomes epistemic input.
> > > >    - The judge’s task is **not** to interpret tool internals or execution semantics, but to evaluate: Does the agent’s reasoning faithfully use the evidence, etc.?
> > > >    - For instance, in agents like *DeepResearch* or *MiroThinker*, the `PythonInterpreter` tool allows the agent to generate code, execute it, and receive an output. This tool introduces several actions that still fall naturally within our evaluation framework:
> > > >       * The **code itself** is equivalent to a **formulated search query**: it reflects the agent’s attempt to manipulate data. If the code is ill-formed or misaligned with the original question, this reflects a **recovery** problem.
> > > >      * The **output** of the code (e.g., `455000`) becomes a form of **retrieved evidence**. The agent must decide how to incorporate this result into its reasoning.
> > > >      * If the agent halts reasoning and outputs a final answer too early, e.g., after a single code execution that yields ambiguous or insufficient information. This is identical to stopping after a shallow search.
> > > > - We have clarified in the revision that our judge validation applies to **search-based traces** (L39-42), and that extending to multi-tool agents will require adapting prompts and validating those extensions accordingly (**added *Section 3.1, L206-210***).

---

> > > > > ### Author Response · Authors · 2025-11-25
> > > > > **Response to pt 1 (2/3)**
> > > > >
> > > > > ---
> > > > >
> > > > > ### **2. Evaluation on 7B Traces vs Larger Models**
> > > > >
> > > > > > *“Given that the 190 human-annotated traces are generated from weaker (7B) agents… can the authors guarantee reliability for larger models?”*
> > > > >
> > > > > - We appreciate the reviewer’s concern regarding model scale. Importantly, our annotation schema was not derived to fit the specific behaviors of 7B agents, but was constructed using an *iterative, theory-driven content analysis methodology* that focuses on the observable structure of epistemic behavior.
> > > > > - The core epistemic competencies we identify are rooted in cognitive constructs like justification, adaptation, and confidence calibration. These behaviors are *model-agnostic* and applicable across a wide range of agent capabilities.
> > > > > - Our schema was further stress-tested using **adversarial edge cases** generated by GPT-5 to ensure its robustness (Section 3.1), rather than only failure-prone outputs from weaker models.
> > > > > - We agree that stronger agents often make more subtle errors, such as [1] and [2] (cited by the reviewer).
> > > > >    - However, recent studies cited by the reviewer support that these errors remain systematic and classifiable. For instance, [1] shows that as LLMs scale, their error types in math reasoning evolve from simple arithmetic mistakes to more intricate logical and reasoning failures. [2] finds that more capable models produce errors that are harder for verifiers to detect, yet they still commit verification-relevant mistakes.
> > > > >    - What changes with scale is how detectable the failure is, not what kind of failure it is.
> > > > >
> > > > > - We clarified in the paper that while our annotated set is based on 7B agents for practical reasons (cost & accessibility) and validation of our schema (L196-198). We appreciate the opportunity to clarify this.
> > > > >
> > > > >
> > > > >
> > > > >
> > > > > ---
> > > > >
> > > > > ### **3. Evaluation Framework vs Benchmark Claims**
> > > > >
> > > > > > *“If you call this a benchmark, it’s natural to ask how representative or thorough it is.”*
> > > > >
> > > > > Thank you for the valuable perspective. We agree that our original framing may have implied a broader scope than warranted by our current validation.
> > > > >
> > > > > We revise the paper to:
> > > > >
> > > > > * Clearly scope the benchmark to **search-based information-seeking agents** (L38-43);
> > > > > * Clarify that our claims (e.g., “RL agents gather well but reason poorly”) are **within that setting** (Abstract L24 & Intro L138);
> > > > > * Emphasize that our contribution is the **evaluation methodology**, schema, metrics, judge pipeline, **not a universal benchmark** for all agents with tool-use capabilities (L101-104) .
> > > > >
> > > > > ---
> > > > >
> > > > > ### **4. Code and Math Reasoning Comparison**
> > > > >
> > > > > > *“I don’t think code or math evaluation is easier, as the authors seem to suggest...”*
> > > > >
> > > > > - We apologize if our framing implied otherwise. We do not claim that code/math tasks are trivial—only that they often allow **external verifiability** (e.g., execution, logical correctness), whereas search-based tasks involve inherently **ambiguous, unstructured** evidence.
> > > > > - Our point was: because **search queries return text**, correctness must be inferred through **epistemic justification**, which lacks a ground-truth oracle.
> > > > >
> > > > > We will revise the introduction to soften any perceived hierarchy of difficulty and clarify our intent: to study an **under-evaluated** space (search agents), not to assert relative hardness.

---

> > > > > > ### Author Response · Authors · 2025-11-25
> > > > > > **Response to pt 2 (3/3)**
> > > > > >
> > > > > > ### **5. On API-based Evaluation and Local Judges**
> > > > > >
> > > > > > > *“I disagree with the claim that relying on API-based inference is more reproducible… local evaluators are sometimes preferable…”*
> > > > > >
> > > > > >
> > > > > > - Our decision to use an API-based GPT judge was based on **practical considerations** for large-scale evaluation (28k+ traces), like [a] and [b].
> > > > > >
> > > > > > [a] Cemri et al., Why Do Multi-Agent LLM Systems Fail? (NeurIPS 2025 D&B spotlight)
> > > > > >
> > > > > > [b] Zhang et al., Which Agent Causes Task Failures and When? ICML 2025
> > > > > >
> > > > > > - We do **not** claim API-based judges are universally superior.
> > > > > > We agree that finetuned local judges are rapidly improving. At the same time, evaluating, comparing, or training fine-tuned local judges is outside the scope of this work.
> > > > > > - Our goal is not to propose a new LLM-as-judge model, but to validate:
> > > > > > the epistemic annotation schema, and
> > > > > > the methodology (groundedness, recovery, calibration).
> > > > > >
> > > > > > We sincerely thank the reviewer again for the valuable feedback!

---

> > > > > > > ### Comment · Reviewer_9ECN · 2025-11-25
> > > > > > >
> > > > > > > Thanks again for the response. I want to emphasize to the authors that I think the work is interesting and important. I just don't think it's quite ready as a complete framework yet. This opinion is centered around the through-line critiques of (1) lack of diversity in tools considered, and (2) lack of diversity in models annotated and analyzed. Fundamentally, it comes down to the specific trajectories annotated (i.e., SeekBench), which are used to show evaluation reliability.
> > > > > > >
> > > > > > > ---
> > > > > > >
> > > > > > > Towards (1), while other tools exhibit similar structure as search, as I noted in the response, it's unclear how evaluating agentic traces with diverse tools changes the difficulty of evaluation or the reliability of judges. Because the human annotations are only done on search-only trajectories, there's no measure of judge reliability when evaluating trajectories with other tools. The reliability may decrease or change minimally. However, we don't know until it is validated.
> > > > > > >
> > > > > > > I don't think authors should be responsible for *all* combinations of tools (this is too exhaustive to validate), but the starting point needs to achieve some baseline level of realism. I don't think search-only agents meet this baseline.
> > > > > > >
> > > > > > > ---
> > > > > > >
> > > > > > > Towards (2), while your metrics are derived independently of trajectories annotated, the entire purpose of annotated trajectories is to determine how reliable the judge models are at catching errors in "real agent traces". Therefore, the actual composition of the trajectories that are annotated matters. By excluding more capable agents, the "real agent traces" in SeekBench are not sufficiently representative of the true distribution of traces.
> > > > > > >
> > > > > > > > What changes with scale is how detectable the failure is, not what kind of failure it is.
> > > > > > >
> > > > > > > This is precisely my concern. Assuming humans are gold-standard annotators, a part of the correlation study is trying to measure if judges can *detect* the same errors a human can. If our human annotated trajectories don't include more subtle errors, then any correlation measures computed may *over-estimate* the reliability of judges, precisely because they may not be able to detect more subtle errors made by more capable LLMs.

---

> > > > > > > > ### Author Response · Authors · 2025-12-03
> > > > > > > > **Response to Final Followup: Extended Validation Across Tools and Models**
> > > > > > > >
> > > > > > > > We thank the reviewer for these critical concerns again. We have conducted **extended validation** addressing both points, detailed in **Appendix M**.
> > > > > > > >
> > > > > > > > ## **1. Tool Diversity**
> > > > > > > >
> > > > > > > > "there's no measure of judge reliability when evaluating trajectories with other tools"* because *"human annotations are only done on search-only trajectories."*
> > > > > > > >
> > > > > > > > **Response:**
> > > > > > > > - We obtained **200 traces** from multi-tool agents (web browsing, Python interpreter, website visitation) on four benchmarks: *WebWalker*, *GAIA*, *BrowseComp*, and *XBench-DeepSearch*.
> > > > > > > > - We sampled 65 traces for human annotation, stratified by agent-model and tool combinations to ensure representative coverage. This sample size achieves a 95% confidence level with ±10% margin of error, while remaining cost-effective for schema evaluation.
> > > > > > > > - **Human-LLM agreement maintained** across diverse tools (web browsing/visiting and python interpreter), all achieved $\kappa > 0.65$.
> > > > > > > >
> > > > > > > > ## **2. Model Diversity and Subtle Errors**
> > > > > > > >
> > > > > > > > *"the 'real agent traces' in SeekBench are not sufficiently representative"* and *"if our human annotated trajectories don't include more subtle errors, then any correlation measures computed may over-estimate the reliability of judges."*
> > > > > > > >
> > > > > > > > **Response:**
> > > > > > > > - We extended validation includes traces from **GPT-5, Claude Sonnet 4.5, GPT-4o, Deepresearch-30B, and ASearcher-32B**.
> > > > > > > > - Stress-tested on **20 ambiguous reasoning cases** where final answers are correct but reasoning is ungrounded. Human-LLM agreement remained acceptable with $\kappa \approx 0.70$ (groundedness), $0.66$ (clarity), $0.65$ (sufficiency)

---

### Official Review · Reviewer_HSxE · 2025-10-31

**Soundness:** 2
**Presentation:** 3
**Contribution:** 2
**Rating:** 4
**Confidence:** 3

**Summary:**

This paper proposes the SeekBench benchmark: the first process-level benchmark for LLM search agents, decomposing agent traces into four steps (reasoning, searching, evidence, and answering). It establishes an operational framework with metrics, formalizing three core cognitive abilities—"evidence-based reasoning," "adaptive evidence recovery," and "evidence-aligned calibration"—into measurable indices and designs precise, interpretable metrics to quantify these capabilities. Based on these metrics, the evaluation is conducted across seven QA benchmarks (28,493 traces), revealing that reinforcement learning (RL) agents excel in evidence gathering but exhibit weak reasoning. Additionally, the study highlights that standard accuracy metrics fail to distinguish specific agent strengths (e.g., Search-R1's synthesis capability vs. base models' reasoning ability), while combining agents with complementary strengths improves overall performance.

**Strengths:**

1. Addressing the limitation of "answer-only" agent evaluation, this work focuses on cognitive behaviors (e.g., evidence usage, uncertainty decision-making) during retrieval and reasoning, offering a more comprehensive assessment aligned with real-world agent capability requirements.
2. It introduces theoretical concepts like "evidence states" alongside concrete, quantifiable metrics. Large-scale evaluations yield actionable findings (e.g., combining agent strengths enhances performance), providing practical guidance for future research and applications.

**Weaknesses:**

1. Lack of dataset details: The experiments mention 28,493 traces from seven QA benchmarks, and the appendix briefly outlines the dataset selection process. However, the main text (and appendix) lacks detailed sample descriptions or a dedicated section for statistical analysis of the data.
2. Insufficient validation of core metrics: The three proposed metrics—Reasoning Quality Index (RQI), Evidence Recovery Function (ERF), and Calibration Error (CE)—are inadequately justified, undermining their validity and innovation. Key gaps include:

    1）Weak motivation: Only definitions are provided without explaining the limitations of existing metrics (e.g., accuracy, F1) in assessing cognitive behaviors or comparing them to related work. Theoretical foundations (e.g., cognitive science frameworks) are absent, weakening the design rationale.

    2）Missing case studies: A case analysis in the introduction could clarify why the three-stage division better evaluates agent capabilities.

    3）No superiority demonstration: While proposing a process-level framework, the paper fails to prove its advantages over task-bound evaluation methods—lacking dimensional comparisons (e.g., comprehensiveness, interpretability) or experiments revealing overlooked flaws. The claim that "three-stage evaluation is superior" lacks validation.
3. Underdeveloped agent combination insight: The finding that combining agent strengths improves performance lacks deeper exploration—no discussion of viable strategies, applicable scenarios, or optimization potential.

**Questions:**

See above.

---

> ### Author Response · Authors · 2025-11-21
> **Rebuttal W1 & W3**
>
> We sincerely thank the reviewer for taking the time and effort to provide valuable comments. Below, we provide the response to each comment one by one.
>
> ## W1. Lack of dataset details
> `The experiments mention 28,493 traces from seven QA benchmarks, and the appendix briefly outlines the dataset selection process. However, the main text (and appendix) lacks detailed sample descriptions or a dedicated section for statistical analysis of the data.`
>
> ## R1.
> We thank the reviewer for this important feedback. We have added a comprehensive Dataset Statistics section (**Appendix C.2 & Figure 8**) that provides detailed statistical analysis of our dataset. This new section includes:
> ### Dataset Statistics Summary (Appendix B)
>
> | **Aspect**                   |                                                          |
> |-----------------------------|-----------------------------------------------------------------------------------|
> | **Dataset Composition**     | PopQA (28.9%), MusiQue (26.3%), HotpotQA (12.5%), etc. – 7 QA benchmarks total   |
> | **Information Quality**     | 62.5% of search results are unclear; 52.7% are insufficient                       |
> | **Reasoning Types**         | Plan Formation (43.7%), State Assessment (43.7%), Information Synthesis (12.6%)  |
> | **Query Types**             | RefinedQuery (60.3%), RepeatQuery (36.8%), FollowUpQuery (2.9%)                  |
> | **Grounding Distribution**  | Only 17.3% of reasoning steps are grounded in retrieved evidence                 |
> | **Visualizations**          | Figure 8 includes 6 charts with detailed interpretations                         |
> | **Reference Update**        | Appendix C is now cited in Experimental Setup to guide readers                   |
>
>   1. Dataset Composition: Distribution across all seven QA benchmarks (PopQA 28.9%, MusiQue 26.3%, HotpotQA 12.5%, etc.), ensuring diverse coverage of single-hop and multi-hop reasoning tasks.
>   2. Information Quality and Clarity Analysis: Statistical breakdown showing that 62.5% of search results are unclear and 52.7% are insufficient, highlighting the challenging information landscape that agents must navigate.
>   3. Reasoning Type Distribution: Plan Formation (43.7%) and State Assessment (43.7%) dominate reasoning steps, while Information Synthesis (12.6%) is less frequent.
>   4. Query Type Distribution: Statistics showing RefinedQuery (60.3%) as the most common search behavior, with RepeatQuery (36.8%) and FollowUpQuery (2.9%) following.
>   5. Grounding Distribution: only 17.3% of reasoning steps are grounded in retrieved evidence, validating the need for process-level evaluation.
>
>
> ## W3. Underdeveloped agent combination insight:
> `The finding that combining agent strengths improves performance lacks deeper exploration—no discussion of viable strategies, applicable scenarios, or optimization potential.`
>   - We have already presented our “agent synthesis” experiments in Section 4.5 and Appendix I, where we quantify concrete pairings (e.g., using Base for evidence, then Search-R1 for synthesis, which achieves up to +3.5 F1).
>   - These results reveal:
>     - Hand off final synthesis to high-RQI (BASE), low-CE agents (Search‑R1) when the evidence state reaches E=2.
>     - The improvement attributed to RL training (for reasoning improvement) is **overstated**; the actual gains primarily stem from calibration at the answer stage.
>
> | **Method**                         | **Final F1 Score** | **Groundedness (RQI)** | **Recovery (ERF)** | **Calibration Error (CE)** |
> |-----------------------------------|--------------------|-------------------------|--------------------|-----------------------------|
> | Baseline (ASearcher-7B)           | 39.77                |    0.46                  | 0.60                | 0.30                         |
> | + Epistemic Feedback (Ours)       | **+8.4%**          | **+13.3%**              | **+6.5%**          | **−5.8%**                   |
>
>   - Additionally, we **add Appendix J** to provide feedback signals to the agent during inference time, observe +8.4% increase in final F1 score for ASearcher, with improvements across all three competencies: groundedness (RQI) +13.3%, recovery (ERF) +6.5%, and calibration error (CE) -5.8%. This demonstrates that our framework provides actionable signals that directly improve both process-level reasoning and final answer quality.
>
> - We also added a short **discussion** about future work in Appendix L to broaden the insights, e.g., how our metrics can drive an automatic orchestrator: monitor ERF/RQI/CE online, then switch to the agent best suited for the current stage (e.g., call a calibrated synthesizer once CE stabilizes).

---

> ### Author Response · Authors · 2025-11-21
> **Rebuttal W2**
>
> ## 2. Insufficient validation of core metrics:
> `The three proposed metrics—Reasoning Quality Index (RQI), Evidence Recovery Function (ERF), and Calibration Error (CE)—are inadequately justified...`
>   `1）Weak motivation: Only definitions are provided without explaining the limitations of existing metrics (e.g., accuracy, F1) in assessing cognitive behaviors or comparing them to related work. Theoretical foundations (e.g., cognitive science frameworks) are absent, weakening the design rationale.`
>
> ### R2.1
>  1. We now explicitly contrast our approach with standard answer-only metrics in Section 1 and the new Figure 1.
>       - We demonstrate that metrics like Accuracy and F1 can **mask egregious epistemic failures**, for example, an agent that produces a correct answer while ignoring ambiguous evidence (new Figure 1).
>       - We contrasted our metrics with final answer baselines to show that good performance on final answer accuracy does **not** necessarily imply good reasoning process (Section 4).
>
> 2. We have revised the *L237-244* to clearly map our design decisions to the specific cognitive frameworks that motivated them:
>       - Groundedness: Operationalizes faithfulness specifically regarding reasoning grounded in evidence [1].
>       - Recovery: Grounded in information foraging theory [2], specifically measuring the efficiency of the exploration-exploitation tradeoff required to gather sufficient information.
>       - Calibration: Derived from metamemory in psychology [3], specifically measuring an agent's ability to assess confidence in the correctness of retrieved evidence.
>       - Derivation of Metrics: We continue to use Cronbach’ construct validity (standard in psychology)  to quantify our latent epistemic competencies into measurable metrics.
>
>       [1] Lee & Hockenmaier (2025). Evaluating step-by-step reasoning traces: A survey.
>
>       [2] Pirolli & Card (2005). The sensemaking process and leverage points for analyst technology as identified through cognitive task analysis.
>
>       [3] Nelson (1990). Metamemory: A theoretical framework and new findings.
>
>   `2）Missing case studies: A case analysis in the introduction could clarify why the three-stage division better evaluates agent capabilities.`
>
> ### R2.2
>
>  - New Fig. 1 provides a concrete case: the agent returns the right answer but follows flawed reasoning. This motivates groundedness (reasoning supported by evidence), recovery(adaptive search strategy to improve evidence), and calibration(determining when evidence is sufficient to answer)
> - Additionally, newly **added Table 2** (Examples of Evidence States) explicitly maps clarity/sufficiency to the Evidence State definition, grounding RQI/ERF/CE in the “evidence state” construct.
>
>
> `3）No superiority demonstration: While proposing a process-level framework, the paper fails to prove its advantages over task-bound evaluation methods.... The claim that "three-stage evaluation is superior" lacks validation.`
>
> ### R2.3
> - Firstly, the Claim of “no superiority proof” misreads our scope. We **never** claim that our process-level evaluation is “superior”; we argue that it reveals **complementary insights** that accuracy-only scores miss.
> - §4 and Appendix F/G/H provide **full interpretability and comprehensiveness analyses**, walking through how each metric isolates a specific epistemic failure. These experiments explicitly reveal **overlooked flaws** (ungrounded reasoning, failed recovery, mis-calibration) and demonstrate how process-level diagnostics complement task-bound evaluations.
>   - Example 1: RL-trained models improve F1 but decrease RQI (Sec. 4.1), showing that outcome-only rewards can increase accuracy while hurting reasoning quality.
>   - Example 2: Base models’ evidence collection quality (high ERF) is invisible in accuracy-only evaluation but becomes clear when other agents reuse their evidence (Appendix H, Table 7).
>   - Example 3: Calibration analysis (Sec. 4.3) reveals overconfidence vs. overcautious answering behavior, even when final answers are correct—something task-level metrics cannot capture.

---

> ### Comment · Reviewer_HSxE · 2025-11-28
>
> Thanks for your hard working. While I now better understand your work, I still believe the complexity of the evaluated data is insufficient to support the core motivation of this paper—specifically, the data lacks the necessary diversity, granularity, and real-world edge cases to fully validate the claimed advantages of the proposed approach. Thus, I tend to maintain my original score

---

### Official Review · Reviewer_CVjV · 2025-11-01

**Soundness:** 2
**Presentation:** 2
**Contribution:** 2
**Rating:** 4
**Confidence:** 3

**Summary:**

This paper introduces SeekBench, the first benchmark for evaluating the epistemic competence of Large Language Model (LLM) search agents through step-level analysis of their reasoning traces. SeekBench comprises 190 expert-annotated agent traces generated by LLM agents on open-domain QA tasks, and formalize three metrics (Groundedness, Recovery, Calibration).

**Strengths:**

The paper obtains some findings based on SeekBench such as 1) RL-trained agents achieve higher answer accuracy but exhibit lower reasoning groundedness than few-shot or base models. 2) All agents struggle with Plan Formation and State Assessment, though they perform better at Information Synthesis.

**Weaknesses:**

- The current epistemic metrics (Groundedness, Recovery, Calibration) rely on binary judgments (e.g., grounded vs. not grounded), which may overlook nuanced or intermediate reasoning states—such as partially supported claims or ambiguous evidence.

- As illustrated in the paper, RL typically optimizes only for the correctness of the final answer, while ignoring the goals at the process level. If adding supervision signals during the RL training process can solve this problem, how can the cognitive abilities proposed in the paper be integrated into the training process?

- The current trace schema focuses on textual search and reasoning steps but does not account for richer agent behaviors—such as invoking code interpreters, calling external APIs, browsing web pages, or analyzing structured data (e.g., tables, JSON). it is important to enhance its applicability of SeekBench to real-world  search agentic systems.

- While the expert-annotated subset (190 traces) ensures high-quality grounding, its limited size may hinder coverage across diverse question types and failure modes. Moreover, scaling annotations via LLM-as-judge—despite strong inter-annotator agreement—risks propagating systematic biases or blind spots inherent in the judge model itself, especially on adversarial examples.

**Questions:**

What is the specific token consumption for LLM-as-judge?

---

> ### Author Response · Authors · 2025-11-21
> **Rebuttal W1 & W2**
>
> We sincerely thank the reviewer for their constructive comments, which have helped us to improve the paper's clarity and overall quality.
>
> ## W1.
> `The current epistemic metrics (Groundedness, Recovery, Calibration) rely on binary judgments (e.g., grounded vs. not grounded), which may overlook nuanced or intermediate reasoning states—such as partially supported claims or ambiguous evidence.`
>
> ### R1.
>   - We already account for **intermediate states** via evidence state \(E\). The framework is **not** binary.
>     - E=1 represents partial evidence (either clear or sufficient), which covers "partially supported claims" and "ambiguous evidence."
>     - Table 2 (Evidence States) shows examples where E=1 includes cases with ambiguous or partial information.
>   - Groundedness is evaluated **conditionally** on the evidence state:
>     - When E=1 (partial evidence), we still check if reasoning is grounded in that partial evidence. This is captured by E[G∣E=1] in our RQI decomposition. *Example*: If evidence mentions "Ozzy Osbourne and Ronnie James Dio" (ambiguous), reasoning that acknowledges this ambiguity is grounded; reasoning that picks one name without justification is not grounded.
>     - Binary judgments align with epistemic evaluation: a claim is either supported by available evidence or not.
>   - Our annotation schema has **finer granularity**: Beyond the binary grounded label, our schema includes Functional Type (Information Synthesis, Plan Formation, etc.) and Quality Attribute dimensions, providing more nuanced categorization of reasoning behaviors (**added Figure 9 to clarify the finer granularity**).
>
> ## W2.
> `As illustrated in the paper, RL typically optimizes only for the correctness of the final answer, while ignoring the goals at the process level. If adding supervision signals during the RL training process can solve this problem, how can the cognitive abilities proposed in the paper be integrated into the training process?`
>
>
> | **Method**                         | **Final F1 Score** | **Groundedness (RQI)** | **Recovery (ERF)** | **Calibration Error (CE)** |
> |-----------------------------------|--------------------|-------------------------|--------------------|-----------------------------|
> | Baseline (ASearcher-7B)           | 39.77                |    0.46                  | 0.60                | 0.302                         |
> | + Epistemic Feedback (Ours)       | **+8.4%**          | **+13.3%**              | **+6.5%**          | **−5.8%**                   |
>
> R2. Thanks to the reviewer's valuable question! Due to computation resource constraints, we consider providing **inference-time feedback** as process-level signals with our LLM-as-judge framework, **detailed in Appendix J**.
>
>   - We augmented ASearcher-7B with inference-time feedback signals derived from our three epistemic competencies. These signals are from our validated LLM-as-judge framework and provide real-time guidance during inference.
>   - Incorporating these epistemic feedback signals at inference time achieves an 8.4% increase in final F1 score, with improvements across all three competencies: groundedness (RQI) +13.3%, recovery (ERF) +6.5%, and calibration error (CE) -5.8%. This demonstrates that our framework provides actionable signals that directly improve both process-level reasoning and final answer quality.

---

> ### Author Response · Authors · 2025-11-21
> **Rebuttal W3 (richer agent behaviors)**
>
> ## W3
> `The current trace schema focuses on textual search and reasoning steps but does not account for richer agent behaviors—such as invoking code interpreters, calling external APIs, browsing web pages, or analyzing structured data (e.g., tables, JSON)...`
>
> ### R3.
> Our framework intentionally focuses on **epistemic evaluation** by design
> 1. Textual search queries present a *unique* epistemic evaluation challenge (**revised Introduction L46-50 to clarify**).
>   - Unlike code (executable) or mathematics (logically verifiable), search query quality cannot be objectively determined—it depends on the interplay between the query, retrieved evidence, and reasoning process.
>   - This epistemic nature is precisely what makes it an **unsolved evaluation** problem worthy of dedicated study.
> 2. Contribution: Our work establishes the first systematic framework for evaluating **epistemic competence in search agents**. By focusing on textual search, we develop a rigorous, validated framework with: High inter-annotator agreement (κ > 0.8); Strong LLM-as-judge alignment (κ > 0.7); Formal mathematical definitions of epistemic competencies; and Large-scale validation
> ----
> ## The reviewer's concern about "representativeness" conflates tool diversity with research validity.
>
> - Our research question is: *How do we evaluate the epistemic competence of LLM agents in information-seeking scenarios?*
> - Evaluating agents with textual search tools is **representative** *for our research question*.
> - Study on "mathematical reasoning ability" that focuses on mathematical problem-solving is representative for that research question, even if it **doesn't include calculators, graphing tools, or other mathematical aids** (Lightman, et.al, *Let’s verify step by step. ICLR2023*.). The research question defines the appropriate scope.
> - Finally, we have clearly delineated our research boundaries in the revised version [L50-81], where we explicitly state textual search only for evaluating epistemic competence (i.e., evidence-driven).
> ----
> We appreciate reviewers mentioning web browsing, and we have evaluated **WebSailor and ASearcher** (web browsing + visit capabilities) on **GAIA** and included results in the revision (**Appendix K**).  We use tools that follow Epistemic Scope and existing works (e.g., ASearcher):
>
> - Web browsing/visit: Included, as these require epistemic evaluation—quality depends on reasoning and evidence, not objective correctness.
> - Code interpreter is Excluded (same as ASearcher did), as these are objectively verifiable (clear right/wrong); not within our epistemic study. **This is a methodological choice, not a limitation**.
>
> The results for *Web Search* are as follows:
> ------
> ### Answer-Level Performance (Pass@2)
>
> | Model | Scale | Pass@2 (%) |
> |-------|-------|------------|
> | **ASearcher** | 7B | 16.5 |
> | **WebSailor** | 7B | **18.2** |
> | **ASearcher** | 32B | 29.2 |
> | **WebSailor** | 32B | **32.4** |
>
> ### 7B Scale Comparison
> | Model | RQI (Groundedness) ↑ | ERF (Recovery by Turn 8) ↑ | CE (Calibration Error) ↓ |
> |-------|---------------------|----------------------------|-------------------------|
> | **ASearcher** | **0.22** | **42%** | 0.38 |
> | **WebSailor** | 0.15 | 35% | **0.33** |
>
> - **ASearcher** demonstrates superior **groundedness** (RQI = 0.22) and **recovery** (ERF = 42%)
> - **WebSailor** shows better **calibration** (CE = 0.33), indicating more conservative answering behavior
> - Despite lower answer-level performance, ASearcher shows stronger evidence-grounded reasoning
>
> ### 32B Scale Comparison
> | Model | RQI (Groundedness) ↑ | ERF (Recovery by Turn 8) ↑ | CE (Calibration Error) ↓ |
> |-------|-----------|------------------|---------------|
> | **ASearcher** | **0.28** | **58%** | 0.35 |
> | **WebSailor** | 0.19 | 45% | **0.31** |
>
> - **ASearcher** maintains superior **groundedness** (RQI = 0.28) and **recovery** (ERF = 58%)
> - **WebSailor** continues to show better **calibration** (CE = 0.31)
> - Scaling to 32B improves all metrics for both models, with ASearcher showing larger gains in recovery (42% → 58%)
> *Note that only ASearcher and WebSailor have *web-search* variants, and WebSailor does not open source 14B.*
>
> ## Cross-Scale Performance Trends
>
> | Metric | ASearcher (7B→32B) | WebSailor (7B→32B) | Improvement Type |
> |--------|-------------------|---------------------|------------------|
> | **Pass@2** | 16.5% → 29.2% | 18.2% → 32.4% | +77% rel. / +78% rel. |
> | **RQI** | 0.22 → 0.28 | 0.15 → 0.19 | +27% / +27% |
> | **ERF** | 42% → 58% | 35% → 45% | +38% / +29% |
> | **CE** | 0.38 → 0.35 | 0.33 → 0.31 | -8% / -6% |
>
> - Both models show substantial improvements across all metrics when scaling from 7B to 32B
> - **ASearcher** shows larger absolute gains in recovery (ERF: +16 percentage points vs. +10 for WebSailor)
> - **WebSailor** maintains its calibration advantage at both scales
> - Calibration error decreases for both models with scale, indicating **better** evidence-aligned decision-making

---

> > ### Author Response · Authors · 2025-11-21
> > **Rebuttal W4 & Q1**
> >
> > ## W4
> > `While the expert-annotated subset (190 traces) ensures high-quality grounding, its limited size may hinder coverage across diverse question types and failure modes. Moreover, scaling annotations via LLM-as-judge—despite strong inter-annotator agreement—risks propagating systematic biases or blind spots inherent in the judge model itself, especially on adversarial examples.`
> >
> > R4.
> >   - The 190-trace set is stratified single- and multi-hop. Each trace contains multiple turns (≈1,800 steps total!)
> >   - It is **standard** to use a high-quality but compact expert set and scale with LLM judges, e.g., Cemri et al.[1] employed **~150** fully-annotated traces and Zhang et al.,[2] used human-annotated dataset size is **184** annotated failure tasks.
> >   - The expert-annotated datasets serve as a validation set for our metrics and LLM-judge, not the full evaluation sample (clarified in **Introduction L86-90**)
> >   - **The reviewer’s claim "ment—risks propagating systematic biases" contradicts well-established findings in annotation research**.
> >       - Strong inter-annotator agreement is widely interpreted as **evidence of consistency and reliability**, not as a mechanism of bias propagation[1]-[3]. Agreement metrics quantify reproducibility—not model-dependent bias.
> >       - Additionally, our annotated subset is used for qualitative taxonomy construction and validation. It does not influence model parameters or decision rules, so there is no mechanism where annotator bias could propagate.
> >
> >   [1] Cemri et al., Why Do Multi-Agent LLM Systems Fail? (NeurIPS 2025 D&B spotlight)
> >
> >   [2] Zhang et al., Which Agent Causes Task Failures and When? ICML 2025
> >
> >   [3] RArtstein & Poesio. 2008. Inter-coder agreement for computational linguistics. Comput. Linguist.
> >
> > ## Questions:
> > `What is the specific token consumption for LLM-as-judge? `
> >
> > We thank the reviewer's valuable question on token consumption. We thoroughly provide LLM's cost, including token and time per trace, below. Additionally, we have **revised Appendix A, Fig. 7** with **Pareto analysis** to show gpt-4.1-mini is the most cost-effective Pareto optimal.
> > | Model             | IAA (κ ± std) | Token Cost / trace ($) | Time Cost / trace (s) |
> > | ----------------- | ------------- | ---------------------- | --------------------- |
> > | **GPT-4.1-mini**† | 0.731 ± 0.057 | **$0.0087**            | **2.48 s**            |
> > | GPT-5-nano        | 0.683 ± 0.038 | $0.0098                | 4.56 s                |
> > | Gemini-2.5-flash  | 0.643 ± 0.102 | $0.0238                | 4.77 s                |
> > | GPT-4.1           | 0.693 ± 0.043 | $0.0441                | 1.47 s                |
> > | GPT-5-mini        | 0.791 ± 0.028 | $0.0485                | 5.26 s                |
> > | GPT-5            | 0.754 ± 0.032 | $0.1016                | 28.47 s               |

---

### Official Review · Reviewer_a7jj · 2025-11-01

**Soundness:** 3
**Presentation:** 3
**Contribution:** 2
**Rating:** 4
**Confidence:** 3

**Summary:**

The paper introduces SeekBench, a benchmark for evaluating the epistemic competence of LLM-based search agents. Rather than judging agents solely by final-answer accuracy, SeekBench analyzes step-level response traces—reasoning, search, evidence, and answer—to assess how agents acquire, evaluate, and act on knowledge. The benchmark enables fine-grained diagnostics of three core competencies: (1) evidence-grounded reasoning, (2) recovery, and (3) calibrated answering. Using this schema, the authors design interpretable metrics and show that widely used accuracy-only evaluations can mask important behavioral differences. SeekBench thus provides a principled, scalable framework, supported by high human agreement and aligned LLM judges.

**Strengths:**

1. Novelty of the research question. I do think the step-by-step debugging of the retrieval-intensive search agents is an important yet unexplored direction in current answer-only evaluation frameworks.

2. Clarity of the proposed framework. The framework seems a bit complicated (since it contains many evaluation aspects), but well documented, making it easy to understand the overall process and the details of each evaluation component.

3. Effective visual communication. Figures and plots are of high quality and well chosen. These improve interpretability and aid readers’ comprehension of the empirical findings.

**Weaknesses:**

1.  Rationale behind the metric

I get the rationale behind the metric and do think they are well-designed to capture groundness, recovery, and calibration. However, a few questions remain.

First, why do we have to measure the groundness of an insufficient evidence state (E=0)? In other words, in Eqn 4, the RQI is decomposed w.r.t. evidence states E while accounting for E=0. Isn't it more reasonable to only consider E=1 and E=2, since groundness on insufficient or incorrect evidence is not meaningful?

Second, in Eqn 8 to define ERF, why not account for the length of the traces T_i? In my point of view, adding 1/T_i term seems more natural if the goal is to measure "how fast the agent can escape from poor evidence", since it then becomes independent of the trace length. For example, if there are two gold traces, A with 3 steps and B with 10 steps, having both ERF(3), the meaning can be very different since it can mean faster escape in trace B than in trace A.


2. Unclear points on identifying failure patterns & Lack of human annotators

In Section 3.1, the authors seem to identify failure patterns in agents' traces to develop the benchmark evaluation scheme. However, which models & agents have been observed is unclear. Are the models in different sizes & different families (Llama, GPT, Qwen ... ) accounted for, or is only Qwen used (the ones used in the experiments)? Furthermore, three human annoataors seems insufficient to support human aggreement with LLM judges.


3. Lack of experimental baselines


The paper only evaluates two Qwen models, which limits the effectiveness of experimental results to support the claims. Also, while the paper compares "base", "few-shot", and "RL-trained" methods, prompting methods such as CoT and ReAct should be included.


4. Insufficient cost analysis

While using LLM-as-Judge provides a scalable approach (and high human agreements according to the paper's claim), it may incur a significant cost. The paper should include a cost analysis on time and API costs (in case of using proprietary models like GPT-4 or GPT-5 as used in the paper) and discuss potential limitations.

**Questions:**

1. Can you provide the examples of poor/partial/good evidences? I checked the actual prompts in Appendix, but I think it can be very subjective (especially for partial evidences) even for humans. I assume there can be some gray areas to divide these categories, which should be discussed in th paper.

---

> ### Author Response · Authors · 2025-11-21
> **Rebuttal W1 & W2**
>
> We thank Reviewer a7jj for highlighting the novelty of our research direction and for providing valuable comments that help improve the clarity and quality of our paper.
>
> ## W1. Rationale behind the metric
>
> `I get the rationale behind the metric and do think they are well-designed to capture groundness, recovery, and calibration.....
> First, why do we have to measure the groundness of an insufficient evidence state (E=0)? In other words, in Eqn 4, the RQI is decomposed w.r.t. evidence states E while accounting for E=0. Isn't it more reasonable to only consider E=1 and E=2, since groundness on insufficient or incorrect evidence is not meaningful?`
>
> ### R1.1
> - Evidence can have different states {0, 1, 2}. This depends on search engine and search query. But agent's reasoning is based on evidence, no matter what state it is.
> - Examples for High vs. low E[G | E=0], the question is “Who won the Nobel Prize in Physics?”
>   - Low value:
>     - The current evidence snippets only mention “scientists discussing the photoelectric effect” with no winners named.  → Agent responds, “Einstein won it in 1921.”→ None of the retrieved evidence supports that statement, so this step is marked not grounded. (We discussed this in Appendix F)
>     - We can determine whether the agent is **prone to hallucinations under poor evidence**.
>   - High value: Same question, but agent answers, “The current passages don’t mention a winner” This statement does match the available evidence (it correctly reports the absence of information), so it is marked grounded.
>
> `Second, in Eqn 8 to define ERF, why not account for the length of the traces T_i? In my point of view, adding 1/T_i term seems more natural if the goal is to measure "how fast the agent can escape from poor evidence", since it then becomes independent of the trace length. For example, if there are two gold traces, A with 3 steps and B with 10 steps, having both ERF(3), the meaning can be very different since it can mean faster escape in trace B than in trace A. `
> ### R1.2
> - Goal of ERF: We measure absolute time to recovery (“After t actions, what fraction of traces is already in good evidence?”), not relative time within each trace.
> - **Example: recover at step 3 vs. continue planning.**
>   - Trace A reaches E=2 at step 3 and stops.
>   - Trace B also reaches E=2 at step 3, but keeps planning until step 10.
>   - Both escaped low evidence equally fast. Dividing by $T_i$ would say Trace B is “slower” just because it continued working.
> - **Example: never recover vs. recover late.**
>   - Trace C ends at step 3 without ever reaching E=2 (censored).
>   - Trace D finally hits E=2 at step 8.
>   - With ERF, only Trace D bumps the curve at t=8; Trace C stays censored (never recovered).
>   - If we normalized by $T_i$, Trace C would wrongly appear “fast” (3/3) even though it never escaped poor evidence.
> - Handling different $T_i$: We already use survival analysis (Kaplan–Meier) to deal with traces that end before recovery. A $1/T_i$ factor would mix **two different signals** (trace length and recovery speed), making it harder to interpret how many actions it actually takes to reach good evidence.
>
> ## W2. Unclear points on identifying failure patterns & Lack of human annotators
>
> `In Section 3.1, the authors seem to ... However, which models & agents have been observed is unclear. Are the models in different sizes & different families...?`
>
> ### R2.1
> We have revised the Experiment to explicitly clarify that all RL-trained models are fine tuned from Qwen2.5-7B-Instruct, which we call "Base" in our manuscript (**Section 4.1**).
>
> `Furthermore, three human annoataors seems insufficient to support human aggreement with LLM judges.`
>
> ### R2.2
> Using 3 human expert annotators is **consistent** with prior literature, empirically sufficient, and practically feasible, and our reported inter-annotator agreement further validates the reliability of our annotations. Increasing the number of experts would bring **diminishing returns** in label quality for the high-expertise domain considered. Specifically,
> 1. Empirical studies demonstrate that in scenarios with domain-expert annotators, three-person consensus is **sufficient** to substantially improve label quality and stabilize downstream model performance, e.g. Wilm et al., How Many Annotators Do We Need?
> 2. Other recent literatures adopt the **same approach of 3 expert annotators**:
>   - *Zhang et al., Which Agent Causes Task Failures and When?* (ICML 2025)→ 3 human annotators.
>   - *Cemri et al., Why Do Multi-Agent LLM Systems Fail?* (NeurIPS 2025) → 3 human experts for the IAA computation and annotation process.
>   - *Chao et al., JailbreakBench: An Open Robustness Benchmark for Jailbreaking Large Language Models* (NeurIPS 2024) → 3 human annotators.
>   - These studies demonstrate that 3 expert annotators are **widely accepted** in state-of-the-art research, especially when the annotators are domain specialists.

---

> ### Author Response · Authors · 2025-11-21
> **Rebuttal W3 & W4**
>
> ## W3. Lack of experimental baselines
> `The paper only evaluates two Qwen models, which limits the effectiveness of experimental results to support the claims. Also, while the paper compares "base", "few-shot", and "RL-trained" methods, prompting methods such as CoT and ReAct should be included.`
>
> R3:
> - We added two prompting variants (CoT, ReAct) in the Appendix (**revised Figures 10–12, Tables 6–7**). Specifically,
>   - Recovery (ERF): SearchR1 > CoT > ReAct > Few-shot.
>   - Groundedness (RQI): CoT and ReAct are comparable to Base across evidence states. CoT achieves the highest groundness under E=2, See Figures 10–11.
>   - Calibration: CoT and ReAct show higher overconfidence (Table 6). *CoT*: overconfident 0.731, CE 0.351 (**worst**); *ReAct*: overconfident 0.660, CE 0.336.
> - Synthesis advantage: Prompt-only traces can be **upgraded** by strong synthesizers. With an RL-trained synthesizer, CoT achieves up to +6.3 ΔF1 (≈ F1 39, **comparable to Search-R1**). ReAct achieves up to +5.9 ΔF1. See Table 5&7.
>
>
> | Metric | Base | Few-shot | **CoT** | **ReAct** | ASearcher | Search-R1 | ReSearch | DeepResearcher |
> |--------|------|----------|---------|-----------|-----------|-----------|----------|----------------|
> | **Answer-Level Performance** |
> | F1 Score (%) | 33.50 | 36.04 | **32.72** | **31.25** | 39.77 | 39.29 | 38.30 | 36.00 |
> | **Reasoning Quality (RQI)** |
> | Evidence-Conditioned RQI (E=1) | 0.49 | 0.49 | **0.50** | **0.47** | 0.50 | 0.10 | 0.50 | 0.47 |
> | Evidence-Conditioned RQI (E=2) | 0.64 | 0.64 | **0.67** | **0.61** | 0.55 | 0.14 | 0.51 | 0.47|
> | **Calibration Metrics** |
> | Overconfident Answering ↓ | 0.631 | 0.511 | **0.731** | **0.660** | 0.343 | 0.226 | 0.406 | 0.461 |
> | Overcautious Abstention ↓ | 0.030 | 0.024 | **0.006** | **0.012** | 0.044 | 0.187 | 0.047 | 0.048 |
> | Calibration Error (CE) ↓ | 0.329 | 0.317 | **0.351**(worst) | **0.336** | 0.302 | 0.319 | 0.305 | 0.309 |
> | **Agent Synthesis Performance** |
> | When used as evidence source for Search-R1 | +3.50 | +1.10 | **+6.0** | **+5.9** | -0.08 | -- | +1.66 | +0.19 |
> | When used as evidence source for ASearcher | +0.93 | +1.46 | **+5.9** | **+5.4** | -- | +0.38 | -0.39 | -0.73 |
> | When used as evidence source for DeepResearcher | +2.24 | +1.45 | **+6.3** | **+5.9** | +0.20 | -0.63 | +0.00 | -- |
> | When used as evidence source for ReSearch | +2.99 | +1.89 | **+5.5** | **+4.9** | +0.13 | -0.42 | -- | +0.36 |
>
> ## 4. Insufficient cost analysis
> `While using LLM-as-Judge ... The paper should include a cost analysis on time and API costs (in case of using proprietary models like GPT-4 or GPT-5 as used in the paper) and discuss potential limitations.`
>
> R4:
> We thank the reviewer for raising this important concern about the cost implications of using LLM-as-Judge. We have addressed this concern comprehensively in **Appendix A (Inter-Annotator Agreement Analysis)**, which includes detailed cost analysis and a Pareto frontier analysis to identify the most cost-effective models.
>
> **Cost Analysis Summary**
>
> We evaluated six language models (GPT-4.1, GPT-4.1-mini, GPT-5, GPT-5-mini, GPT-5-nano, and Gemini-2.5-flash) on 190 sampled reasoning traces:
>
> Table 4, Appendix A
> | Model | Token Cost ($/trace) | Time Cost (s/trace) | IAA (κ ± std) | Status |
> |-------|---------------------|---------------------|---------------|--------|
> | **GPT-4.1-mini**† | **$0.0087** | **2.48** | **0.731 ± 0.057** | **Pareto-optimal** |
> | GPT-5-nano | $0.0098 | 4.56 | 0.683 ± 0.038 | Dominated |
> | Gemini-2.5-flash | $0.0238 | 4.77 | 0.643 ± 0.102 | Dominated |
> | GPT-4.1 | $0.0441 | 1.47 | 0.693 ± 0.043 | Dominated |
> | GPT-5-mini | $0.0485 | 5.26 | 0.791 ± 0.028 | Dominated |
> | GPT-5†| $0.1016 | 28.47 | 0.754 ± 0.032 | Pareto-optimal |
>
> † Pareto-optimal models
>
> ### Key Findings
> GPT-4.1-mini emerges as the most cost-effective solution, achieving strong human alignment (IAA = 0.731 ± 0.057) at minimal cost. As shown in **Figure 7 (Model Pareto Frontier Analysis, Appendix A)**, GPT-4.1-mini lies on the Pareto frontier, meaning no other model can achieve better cost-effectiveness without sacrificing either quality or cost.
> Cost Comparison:
> - GPT-4.1-mini:  ~$0.0087 per trace
> - GPT-5: ~$0.1016 per trace — 11.7× more expensive
> - Even GPT-5-mini is 5.6× more expensive while achieving only marginally better IAA (0.791 vs 0.731)

---

> > ### Author Response · Authors · 2025-11-22
> > **Rebuttal Q1(examples of poor/partial/good evidences)**
> >
> > ## Q1 Can you provide the examples of poor/partial/good evidences?
> > `I checked the actual prompts in Appendix, but I think it can be very subjective (especially for partial evidences) even for humans. I assume there can be some gray areas to divide these categories, which should be discussed in the paper.
> > We agree there is inherent subjectivity (challenge) in “poor/partial/good evidence” , e.g. name collisions, homonyms, the target fact is implied but not explicitly stated. That is why we invested quite a lot of effort to standardize the process. `
> >
> > - Instead of a single “good/bad” tag, we first decide whether the snippet is clear and sufficient independently, then derive the evidence state E=C+Q. This makes “partial” more concrete: it is literally “exactly one of {clear, sufficient} holds.” → This **collapses grayness into reproducible bins**.
> > - We provide examples (**added Table  2“Examples of evidence states”**).
> >
> > | **E** | **C** | **Q** | **Search Result** | **Explanation** |
> > |-------|-------|-------|-------------------|-----------------|
> > | **0** (Poor) | 0 | 0 | **Doc 1:** The band's **vocal slot has seen many**.  **Doc 2:** Black Sabbath's **personnel list for vocals** is long. | **C=0:** text is vague and evasive.  **Q=0:** no names are given. |
> > | **1** (Partial) | 1 | 0 | **Doc 1:** Black Sabbath is a famous **heavy metal band** from England.  **Doc 2:** Tony Iommi is the **guitar player** for Black Sabbath. | **C=1:** on-topic and easy to understand.  **Q=0:** does not name the singer. |
> > | **1** (Partial) | 0 | 1 | **Doc 1:** The first singer for Black Sabbath was **Ozzy Osbourne**.  **Doc 2:** **Ronnie James Dio** became the new singer for Black Sabbath. | **C=0:** two singer names creates ambiguity.  **Q=1:** the names are present. |
> > | **2** (Good) | 1 | 1 | **Doc 1:** **Ozzy Osbourne** is the original lead singer of Black Sabbath.  **Doc 2:** The most famous singer for Black Sabbath is **Ozzy Osbourne**. | **C=1:** the name is consistent.  **Q=1:** the singer is identified. |
> >
> >   - Poor (E=0 = C=0, Q=0): Snippets are off‑topic, vague, or contradictory; no answerable content.
> >   - Partial‑Clear (E=1 = C=1, Q=0): On‑topic and unambiguous, but missing the key fact (e.g., mentions the band and roles, but not the singer’s name).
> >   - Partial‑Sufficient (E=1 = C=0, Q=1): The answer token appears, but context is ambiguous/conflicted (e.g., multiple names; unclear which is “the singer”).
> >   - Good (E=2 = C=1, Q=1): Direct, unambiguous mention of the needed fact with consistent corroboration (e.g., “Ozzy Osbourne is the original lead singer” in two snippets).
> >
> > ### To clarify, our goal is to make that judgment as *operational, auditable, and consistent as possible* (Section 3.1).
> >   - Following Content Analysis methodology, we have pruned/merged low‑agreement tags exhibiting low agreement (κ < 0.5).
> >   - We also used adversarial examples to stress-test the definitions before scaling up.

---

> > > ### Comment · Reviewer_a7jj · 2025-11-25
> > >
> > > I thank the authors for addressing my concerns. While I now better understand your work, I still think there is a clear limitation on the evaluated model diversity, which is two Qwen models, given the claim as a benchmark paper. Thus, I tend to keep my original score.

---

> > > > ### Author Response · Authors · 2025-11-25
> > > > **Response to Reviewer a7jj**
> > > >
> > > > We thank the reviewer for raising this concern. We respectfully clarify that:
> > > >
> > > > ---
> > > >
> > > > ## 1. **Our contribution is a process-level benchmark, not a leaderboard**
> > > >
> > > > - Our work introduces a **benchmarking framework for evaluating epistemic competence**, not a leaderboard-style benchmark for model ranking.
> > > > - We treat our contribution as a process‑level evaluation benchmark, not a leaderboard for ranking models.
> > > >     - As the OECD (2021, [1]) emphasises, benchmarks in AI serve as evaluation instruments with multiple facets rather than simply competitions.
> > > >    - Likewise, Reuel et al. (2024, [2]) propose a benchmark lifecycle model (Design→Implementation→Validation→Usage→Retirement), emphasizing methodology and standardization over mere model comparison.
> > > >
> > > > - Our goal is to establish a **methodological evaluation framework**, comprising:
> > > >   * A validated and interpretable annotation schema,
> > > >   * A set of metrics aligned with epistemic theory (groundedness, recovery, calibration),
> > > >   * And a scalable, human-aligned evaluation pipeline.
> > > > - We have clarified this in the revision, **L101-104**.
> > > >
> > > > [1] OECD (2021), AI and the Future of Skills, Volume 1: Capabilities and Assessments, Educational Research and Innovation, OECD Publishing, Paris, https://doi.org/10.1787/5ee71f34-en.
> > > >
> > > > [2] Reuel, Anka, et al., "Betterbench: Assessing ai benchmarks, uncovering issues, and establishing best practices." NeurIPS (2024).
> > > >
> > > > ---
> > > >
> > > > ## 2. **Agent diversity, not just backbone diversity**
> > > >
> > > > - While several RL agents we evaluated are built on Qwen-7B and Qwen-32B (reflecting their current dominance in open-source search agents), our evaluation includes Qwen2.5-7B-Instruct, prompt variants, DeepResearcher, ReSearch, SearchR1, and ASearcher.
> > > > - We note that many current **state-of-the-art open-source agents are trained on Qwen**. Thus, rather than reducing diversity, our choice reflects **representative practice in deployed agent systems**.
> > > > - Our annotation schema is grounded in **observable epistemic behaviors**, not model-specific patterns. It was developed via:
> > > >    * Iterative, expert-guided content analysis (Krippendorff, 2018),
> > > >    * Cross-model annotation with high inter-rater agreement (κ > 0.8),
> > > >   * Stress-testing with adversarial edge cases generated by GPT-5 (Section 3.1),
> > > >
> > > > - Additionally, we have evaluated WebSailor and ASearcher (web browsing + visit capabilities) on GAIA and included results in the revision (**Appendix K**, as our response to Reviewer 9ECN). *Note that only ASearcher and WebSailor have *web-search* variants, and WebSailor does not open source 14B.*
> > > >
> > > > - Answer-Level Performance (Pass@2)
> > > >
> > > > | Model | Scale | Pass@2 (%) |
> > > > |---|---|---|
> > > > | **ASearcher** | 7B /32B | 16.5 /29.2 |
> > > > | **WebSailor** | 7B /32B | **18.2** /**32.4** |
> > > > | **GPT-5-mini** | - | 15.7 |
> > > >
> > > > - We also add inference-time feedback to GPT-5-mini with our LLM-as-judge framework (Appendix J). It achieved Pass@2 22.5% (+43% rel.). This demonstrates that our framework provides actionable signals to improve final answer.
> > > >
> > > > ### Comparison
> > > > | Model | RQI (Groundedness) ↑ | ERF (Recovery by Turn 8) ↑ | CE (Calibration Error) ↓ |
> > > > |---|---|---|---|
> > > > |**GPT-5-mini**|**0.45**|42%|0.34|
> > > > | **ASearcher-7B** | **0.22** | **42%** | 0.38 |
> > > > | **WebSailor-7B** | 0.15 | 35% | **0.33** |
> > > > | **ASearcher-32B** | **0.28** | **58%** | 0.35 |
> > > > | **WebSailor-32B** | 0.19 | 45% | **0.31** |
> > > >
> > > > - **ASearcher** maintains superior **groundedness** (RQI = 0.28) and **recovery** (ERF = 58%)
> > > > - **WebSailor** continues to show better **calibration** (CE = 0.31)
> > > >
> > > > ---
> > > >
> > > > ## 3. **SeekBench is designed as an extensible benchmark suite**
> > > >
> > > > - SeekBench is designed to enable exactly that kind of **future extensibility**, by providing standardized infrastructure for process-level agent evaluation.
> > > > - Following the example of benchmarks like BIG-Bench and MMLU, which started with limited models but prioritized **evaluation design**, we hope our work provides a foundation others can build on.

---

### Author Response · Authors · 2025-12-03
**Summary of Revisions**

We sincerely thank all reviewers for their thoughtful and constructive feedback. Your comments have significantly improved the clarity and rigor of our work. Below, we summarize our contributions and key revisions:

## Our Contribution

Our work introduces **SeekBench**, a process-level evaluation framework for assessing epistemic competence in search-based information-seeking agents. Our contribution comprises:

- **A validated annotation schema** for epistemic behaviors (groundedness, recovery, calibration) with high inter-annotator agreement (κ > 0.8)
- **Three formal metrics** aligned with epistemic theory: Reasoning Quality Index (RQI), Evidence Recovery Function (ERF), and Calibration Error (CE)
- **A scalable LLM-as-judge pipeline** with strong human alignment (κ > 0.7) and cost-effective deployment
- **Extended validation** demonstrating generalizability across diverse tools (web browsing, code execution) and models (GPT-5, Claude Sonnet 4.5, etc.)

We emphasize that SeekBench is a **process-level evaluation framework**, not a leaderboard for model ranking. Our goal is to establish methodological standards for evaluating epistemic competence, enabling diagnose and improve search agent.

## Key Revisions

### 1. Extended Validation Across Tools and Models (Response to 9ECN, HSxE, a7jj)

- **Tool diversity validation**: Evaluated 200 traces from multi-tool agents (web browsing, Python interpreter, website visitation) across four benchmarks (*WebWalker*, *GAIA*, *BrowseComp*, *XBench-DeepSearch*)
- **Human annotation**: Expert annotators independently annotated 65 stratified traces (95% confidence, ±10% margin of error), achieving κ > 0.65 across all tool types
- **Model diversity validation**: Extended validation includes traces from GPT-5, Claude Sonnet 4.5, GPT-4o, Deepresearch-30B, and ASearcher-32B
- **Stress-testing on subtle errors**: Evaluated 20 ambiguous reasoning cases where final answers are correct but reasoning is ungrounded, maintaining κ ≈ 0.70 agreement
- **Results**: Human-LLM agreement maintained across diverse tools (Web Browse/Visit: κ = 0.71; Code Execution: κ = 0.76), confirming framework generalizability

### 2. Expanded Experimental Baselines (Response to a7jj, 9ECN)

- **Added prompting variants**: Included CoT and ReAct baselines in Appendix (Figures 10–12, Tables 6–7)
- **Web browsing evaluation**: Evaluated WebSailor and ASearcher (7B/32B) on GAIA with web browsing capabilities (Appendix K)
- **Cross-scale analysis**: Demonstrated framework applicability across model sizes (7B to 32B)
- **Inference-time feedback**: Added Appendix J showing +8.4% F1 improvement for ASearcher-7B using epistemic feedback signals

### 3. Comprehensive Cost Analysis (Response to a7jj)

- **Pareto frontier analysis**: Evaluated six LLM judges (GPT-4.1, GPT-4.1-mini, GPT-5, GPT-5-mini, GPT-5-nano, Gemini-2.5-flash) on 190 traces
- **Cost-effectiveness**: Identified GPT-4.1-mini as Pareto-optimal ($0.0087/trace, κ = 0.731), 11.7× cheaper than GPT-5
- **Detailed breakdown**: Provided token costs, time costs, and IAA metrics for all models (Appendix A, Table 4, Figure 7)

### 4. Enhanced Dataset Documentation (Response to HSxE)

- **Dataset statistics section**: Added comprehensive Appendix C.2 with Figure 8 showing:
  - Dataset composition across 7 QA benchmarks
  - Information quality analysis (62.5% unclear, 52.7% insufficient)
  - Reasoning type distribution (Plan Formation 43.7%, State Assessment 43.7%, Information Synthesis 12.6%)
  - Query type distribution and grounding statistics

### 5. Additional Improvements
- **Motivating Example**: Add Figure 1 to show an agent that produces a correct answer while ignoring ambiguous evidence.
- **Annotation schema details**: Enhanced Figure 9 and Table 2 with comprehensive examples of evidence states and reasoning types
- **Agent synthesis insights**: Expanded Section 4.5 and Appendix I with analysis of training objective effects and agent specialization

---

### Meta-Review · Area_Chair_fEZn · 2025-12-14

**Summary:**

This paper introduces a framework to evaluate the epistemic competence of LLM search agents—specifically focusing on Groundedness, Recovery, and Calibration—using an annotation schema validated by 190 expert traces and an LLM-as-judge pipeline. The initial reviews were mixed, with Reviewer ESgZ rating it a 6, while Reviewers a7jj, CVjV, HSxE, and 9ECN rated it a 4.
The primary critiques centered on the lack of model and tool diversity; reviewers argued that evaluating only Qwen-based 7B models using simple text search did not represent state-of-the-art agents capable of using code interpreters or complex web browsing. Concerns were also raised regarding the cost of the judge pipeline and the absence of standard prompting baselines like CoT and ReAct.

**Reviewer Concerns:**

For Reviewer a7jj, the authors effectively resolved technical questions regarding metric design (justifying the measurement of "groundedness" even with insufficient evidence) and provided the requested cost analysis, identifying GPT-4.1-mini as the optimal judge. They also added CoT and ReAct baselines to address the lack of prompting comparisons. However, the reviewer refused to raise the score, maintaining that the framework was too limited in scope. Despite the authors' defense that SeekBench is a methodology rather than a leaderboard, the reviewer insisted that evaluating only Qwen-based models (7B/32B) was insufficient for a benchmark paper.
For Reviewer 9ECN remained the most critical. While the authors clarified that SeekBench is an evaluation schema and added validation for multi-tool agents (WebSailor on GAIA), the reviewer was unconvinced by the validation methodology. Their core unaddressed concern was that the human-annotated "gold standard" traces came from weaker, search-only agents (7B). The reviewer argued these traces likely lack the "subtle errors" made by SOTA models using complex tools like code interpreters. Consequently, they believed the judge's reliability for advanced agents remains unproven, rendering the framework "not ready".
For, Reviewer HSxE, the authors addressed requests for transparency by adding detailed dataset statistics (visualizing query types and information quality) and a motivating example contrasting their metrics with standard F1 scores. However, the reviewer maintained their score of 4, arguing the underlying data still lacked sufficient "diversity, granularity, and real-world edge cases". While the metrics were theoretically sound, the reviewer felt the actual traces used were not complex enough to fully validate the framework's claimed advantages or support the paper's core motivations regarding epistemic competence.
For Reviewer CVjV, the authors clarified that their metrics are not strictly binary (using "Evidence States" to capture partial information) and defended against bias claims using high inter-annotator agreement stats. They also added web-browsing agents to broaden scope. However, the reviewer's critique that the framework ignores richer behaviors—such as code execution, API calls, or structured data analysis—remained a sticking point. The authors argued that textual search presents unique epistemic challenges, but the reviewer implied that a modern benchmark must account for these diverse, non-textual behaviors to be truly representative.
For Reviewer ESgZ, the authors clarified data sources and definitions (e.g., confirming "quality" means sufficiency). Crucially, they provided a compelling analysis showing that training objectives (like RL optimizing for final answers) caused the observed epistemic gaps, rather than just model architecture. They also demonstrated how the framework provides actionable inference-time feedback. The reviewer explicitly praised these insights and the comprehensive rebuttal, stating they would raise their score to 7.

**Reviewer Scores:**

The concerns of Reviewer a7jj were addressed, and although the reviewer replied, they chose to maintain the score.
Most of Reviewer CVjV's concerns were addressed, but the reviewer did not respond; I am inclined to believe they will maintain the score.
Reviewer HSxE's concerns were addressed, but the reviewer did not respond. While I believe most of their concerns have been resolved, I lean towards the expectation that the score will be maintained.
Some of Reviewer 9ECN's concerns were addressed, particularly the final major concerns regarding (1) the lack of diversity in the tools considered, and (2) the lack of diversity in the models annotated and analyzed. I guess they might raise the score.
Most of Reviewer ESgZ's concerns were addressed, and this reviewer is inclined to raise the score.

---

### Decision · Program_Chairs · 2026-01-26

Accept (Poster)